# The Future Unmarked: Watermark Removal in AI-Generated Images via Next-Frame Prediction

**Huming Qiu[1], Zhaoxiang Wang[1], Mi Zhang[1],*, Xiaohan Zhang[1], Xiaoyu You[2], Min Yang[1],***

[1]Fudan University, Shanghai, China
[2]East China University of Science and Technology, Shanghai, China
{hmqiu23@m., zhaoxiangwang25@m., mi_zhang@, xh_zhang@, m_yang@}fudan.edu.cn
xiaoyuyou@ecust.edu.cn

## Abstract

Image watermarking embeds imperceptible signals into AI-generated images for deepfake detection and provenance verification. Although recent semantic-level watermarking methods demonstrate strong resistance against conventional pixel-level removal attacks, their robustness against more advanced removal strategies remains underexplored, raising concerns about their reliability in practical scenarios. Existing removal attacks primarily operate in the pixel domain without altering image semantics, which limits their effectiveness against semantic-level watermarks. In this paper, we propose *Next Frame Prediction Attack (NFPA)*, the first semantic-level removal attack. Unlike pixel-level attacks, NFPA formulates watermark removal as a video generation task: it treats the watermarked image as the initial frame and aims to subtly manipulate the image semantics to generate the next-frame image, i.e., the unwatermarked image. We conduct a comprehensive evaluation on eight state-of-the-art image watermarking schemes, demonstrating that NFPA consistently outperforms thirteen removal attack baselines in terms of the trade-off between watermark removal and image quality. Our results reveal the vulnerabilities of current image watermarking methods and highlight the urgent need for more robust watermarks. Code is available at `https://github.com/1249748036/NFPA`.

## 1   Introduction

Text-to-image (T2I) generation models [34, 35, 38] have found widespread applications across various domains [11, 36, 23], yet they also face significant risks of malicious misuse, particularly in generating deepfake content [16, 40, 1]. Numerous cases demonstrate that these models are exploited to produce false information [4] and inappropriate materials [10]. As an active defense mechanism, image watermarking embeds imperceptible watermarks into generated images to effectively mitigate the negative consequences of model abuse [39]. For example, DeepMind employs the SynthID [14] to embed invisible watermarks in AI-generated images for copyright protection and misuse prevention. However, since malicious attackers may become aware of these watermarks and attempt to remove them, watermarks must possess sufficient robustness to withstand potential removal attacks [37].

To address this issue, many image watermarking schemes prioritize robustness as a core design principle, with a gradual research shift from pixel-level to semantic-level watermarks that offer greater potential for resilience [48]. For example, HiDDeN [49], an early pixel-level watermarking scheme, embeds watermarks by introducing small perturbations in the pixel space and incorporates noise simulation layers to enhance robustness against noise-based distortions. In contrast, TreeRing [43]

---

*Corresponding authors

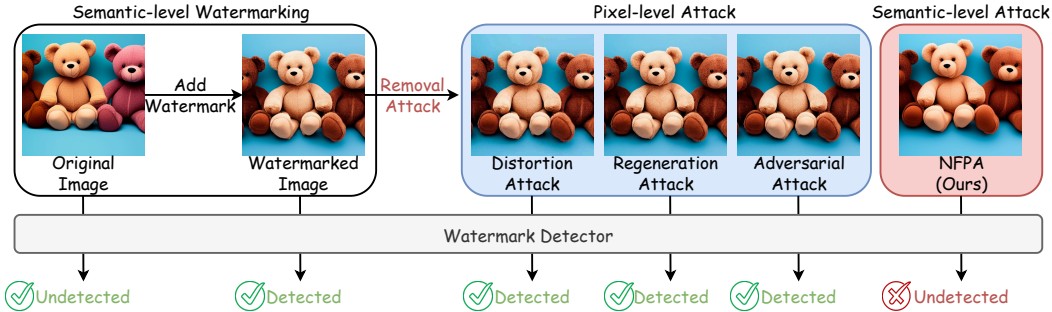

Figure 1: Examples of existing removal attacks. Pixel-level attacks attempt to remove the watermark by modifying pixels, yet the semantic-level watermark remains detectable. In contrast, our attack manipulates image semantics to effectively evade watermark detection.

represents a recent semantic-level approach that embeds circular watermark patterns in the initial noise frequency domain of T2I models, thereby manipulating the semantic features of the generated images. This design improves the watermark's adaptability to transformations such as translation and rotation. Compared to pixel-level watermarks, semantic-level watermarks leverage the semantic structure of images for detection, making them inherently more robust to image transformations, denoising, and other perturbations [8].

However, despite the robustness of state-of-the-art (SOTA) image watermarking schemes against certain removal attacks, their resilience remains insufficiently validated and underexplored. This gap may lead to an overestimation of their reliability in real-world applications, fostering a false sense of security [3, 22]. As illustrated in Figure 1, existing attack methods primarily operate at the pixel level, attempting to remove watermarks through minor perturbations in the pixel space without modifying the underlying semantic structure. This limitation may explain their reduced effectiveness against semantic-level watermarks. Consequently, a natural question arises:

*Is there a semantic-level removal attack capable of removing SOTA image watermarks?*

**Our Work.** This paper, for the first time, proposes a semantic-level watermark removal attack, providing a clear affirmative answer to the question raised above. Unlike traditional pixel-level removal methods, we develop a semantic-level attack that removes watermarks in AI-generated images by manipulating their semantic structure. However, the semantic-level attack faces several key challenges, primarily how to effectively remove the watermark while preserving the semantic content of the image as much as possible (see section 3 for details).

Inspired by advances in video generation [21, 31, 30], we propose Next Frame Prediction Attack (NFPA) to address these challenges. Specifically, NFPA formulates the semantic-level watermark removal task as a next-frame prediction problem: it treats the watermarked image as the initial frame $x_0$, and removes the watermark by semantically modifying the image through the prediction of the next frame $x_1$. Since temporal consistency is a fundamental property of video generation, $x_1$ typically differs only slightly from $x_0$, and the two frames are generally considered visually equivalent [45], thereby ensuring semantic consistency between the attacked and watermarked images. In addition, NFPA possesses several key properties: *(i) universal*, effective against all image watermark types, *(ii) black-box*, requiring no knowledge of the watermark, *(iii) data-free*, requiring no additional data, and *(iv) query-free*, requiring no feedback from the detector.

To enable the effectiveness of NFPA, we design a novel video (frame) generation framework to support our attack pipeline. This architecture leverages a pre-trained T2I generation model and adapts it as a next-frame prediction model in a zero-shot, tuning-free manner, allowing for the rapid generation of high-quality next-frame images without additional training. Specifically, we take the watermarked image as a conditional input and obtain its latent representation through the DDIM inversion process, which serves as the initial-frame noise. We then construct a flow matrix to simulate the motion trajectory of the next frame and accordingly warp the initial noise to produce the noise for the next frame. To effectively remove the watermark, we constrain the next-frame noise to lie within a predefined search space while maximizing its distance from the initial noise. Furthermore, we replace the standard self-attention mechanism with a frame-level attention mechanism to enhance

spatiotemporal consistency between adjacent frames. Finally, we apply the denoising process to generate the next-frame image, which corresponds to the unwatermarked image.

Overall, the main contributions of this paper are as follows:

- We propose the first semantic-level image watermark removal attack, *Next Frame Prediction Attack (NFPA)*, which focuses on removing watermarks by modifying the semantic structure of the image. Drawing inspiration from video generation, we reframe the semantic-level removal attack as a next-frame prediction task, ensuring that the attacked image maintains semantic consistency with the original watermarked image.

- We design a novel zero-shot, tuning-free next-frame prediction framework, which takes the watermarked image as the initial frame condition and efficiently generates unwatermarked images through next-frame prediction. By introducing a flow matrix with a maximization search strategy, NFPA effectively facilitates watermark removal.

- We conduct a systematic evaluation of NFPA on eight image watermarking schemes and compare it with thirteen removal attack baselines. The experimental results validate that NFPA effectively removes SOTA image watermarks while preserving image quality, further revealing significant shortcomings in the robustness of current image watermarking schemes.

## 2 Related Work

### 2.1 AI-Generated Image Watermarks

Watermarking for AI-generated images aims to embed imperceptible watermarks into generated content [26]. These watermarks are nearly invisible to the human eye but can be reliably detected by designated detectors, making them widely applicable in areas such as copyright protection [19, 50, 27] and deepfake detection [29, 47, 6]. Early developments in this field rely primarily on traditional handcrafted methods, such as DwtDct [9], which embed watermarks in the frequency domain using wavelet and discrete cosine transforms. Recent research shifts toward leveraging deep learning models to embed watermarks, with the goal of enhancing watermark robustness. Based on the embedding stage, image watermarking schemes are generally categorized into two types [3]: Post-processing Watermarks, which add watermarks after image generation, and in-processing watermarks, which integrate watermarking during the generation process.

**Post-processing Watermarks.** StegaStamp [41] adopts a joint optimization framework of encoder-decoder to encode and decode the watermark, introducing a noise layer to enhance the robustness of the watermark. RivaGAN [46] improves the encoder-decoder framework by incorporating an attention mechanism. SSL watermarking [12] employs a self-supervised training paradigm, using data augmentation to enhance the watermark's adaptability to image transformations. However, these watermarks are essentially subtle disturbances and are vulnerable to denoising or other more advanced attack methods. Although StegaStamp demonstrates good robustness, it may leave noticeable artifacts in the image [41].

**In-processing Watermarks.** Stable Signature [13] fine-tunes the VAE decoder of the diffusion model to ensure the generated image contains a watermark. Gaussian Shading [44] offers a lossless watermarking solution by mapping the watermark message into latent representations that follow a standard Gaussian distribution. TreeRing [43] introduces preset patterns into the initial noise of the diffusion model, causing a change in the semantic structure of the generated image. RingID [8] improves upon Tree-Ring, optimizing watermark capacity.

In addition, watermarking methods can also be categorized based on whether they alter the semantic content of the image, distinguishing between pixel-level and semantic-level watermarks. The former, such as DwtDct, RivaGAN, SSL, and Stable Signature, typically embed ownership signals directly into the image pixels or frequency domain through perturbations or frequency modulation. These approaches are generally subtle and nearly imperceptible to humans but are vulnerable to common distortions such as compression, cropping, or noise injection. The latter, including TreeRing, RingID, Gaussian Shading, and StegaStamp, typically operate in the latent space of diffusion models, encoding ownership by modifying high-level semantic features. Semantic-level watermarks offer stronger robustness and are considered a powerful alternative to pixel-level watermarks [48], but it may also introduce potential sensitivity to changes in semantic structure.

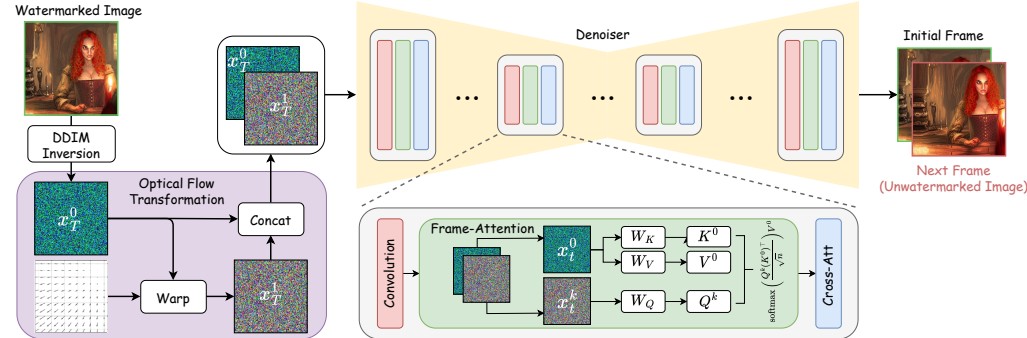

Figure 2: Pipeline of NFPA. By adapting a pretrained T2I model as a next-frame prediction framework, NFPA takes a watermarked image as input and generates a semantically coherent next-frame prediction in which the watermark is effectively removed.

## 2.2 Watermark Removal Attacks

Watermark removal attacks aim to remove embedded watermarks in images to evade watermark detection. According to their underlying attack mechanisms, existing methods can be categorized into three types [3, 20]: distortion attacks, regeneration attacks, and adversarial attacks.

Distortion attacks involve common image processing operations applied during image transmission, such as cropping, rotation, and JPEG compression. These attacks are characterized by their simplicity and efficiency but often achieve only limited effectiveness in removing watermarks [49]. Regeneration attacks seek to remove watermarks by reconstructing the image using generative models. The typical process involves adding noise to the watermarked image and then reconstructing it through models such as diffusion models or variational autoencoders. While regeneration attacks are particularly effective against pixel-level watermarking methods, recent semantic-level watermarking techniques demonstrate significant robustness against such attacks [48]. In contrast, adversarial attacks optimize adversarial perturbations to mislead watermark detectors. However, these attacks require stronger attacker capabilities, such as knowledge of the watermark or access to the watermark detector. Attackers may also collect watermarked and clean images to train surrogate detectors, leveraging transferability to conduct adversarial attacks. The additional attack cost limits their application, though they show some effectiveness against SOTA semantic-level watermarking schemes [37].

## 3 Method

In this section, we present a detailed description of NFPA and explain how it addresses two key challenges associated with semantic-level removal attacks through its methodological design: *(i) watermark removal*, how it ensures that the semantic changes are effective enough to cause watermark detection failure; *(ii) semantic preservation*, how it ensures that semantic modifications maintain visual consistency with the original image, thus preserving perceptual fidelity.

### 3.1 Overview

Motivated by advances in video generation, we formulate the semantic-level image watermark removal problem as a video generation task. However, existing video generation methods either fail to provide effective guidance for watermark removal or incur substantial training and computational costs, making them unsuitable for NFPA. To address this, we design a novel image-to-video generation framework to enhance the targeting and efficiency of NFPA's attack. This framework leverages the generation capabilities of T2I models, such as Stable Diffusion-v2.1-base [2], to perform zero-shot next-frame prediction, enabling watermark removal without any training or fine-tuning. Notably, NFPA naturally benefits from ongoing improvements in image generation models.

Formally, We define an image-to-video generation function $f$, which takes a watermarked image $x \in \mathbb{R}^{H \times W \times 3}$ as input and outputs a sequence of video frames $V \in \mathbb{R}^{m \times H \times W \times 3}$, where $H \times W$

denotes the image resolution and $m$ denotes the number of frames. In our attack setting, we fix $m = 2$ to significantly reduce the computational cost of generation, thereby simplifying $f$ into a next-frame prediction function. As a result, the output video sequence $V$ consists of only two frames, denoted as $V = [x^0, x^1] \in \mathbb{R}^{2 \times H \times W \times 3}$, where $x^0$ represents the reconstruction of the input watermarked image $x$, and $x^1$ denotes the next frame based on $x$. Accordingly, the attack pipeline of NFPA is formalized as $f(x) = [x^0, x^1]$, where $x^1$ serves as the attack target, i.e., the unwatermarked image.

**Attack Pipeline.** Figure 2 illustrates the architecture of our zero-shot next-frame prediction framework and the overall attack pipeline of NFPA. As the first step, NFPA performs DDIM inversion to map the input watermarked image $x$ to its corresponding noise representation $x_T$. DDIM inversion approximates the reverse diffusion process, aiming to find a latent noise $x_T$ such that the forward denoising process approximately reconstructs the original image, i.e., $x \approx \text{Denoiser}(x_T)$. By applying this operation, we extract the noise representation $x_T^0$ from the input watermarked image $x$, which serves as the initialization for the next-frame prediction task. To model the dynamics of video frames, we construct an optical flow matrix based on $x_T^0$ to simulate the spatial motion trajectory of the next frame. Specifically, we apply an optical flow transformation to $x_T^0$, resulting in a candidate next-frame noise representation $x_T^1$. To enhance watermark removal, we constrain the optical flow matrix within a restricted search space and optimize it to maximize the distance between $x_T^1$ and $x_T^0$, thereby disrupting with the potential watermark signal.

To ensure spatiotemporal consistency between adjacent frames, we replace the self-attention module in the diffusion model with a frame-attention module, which better captures semantic dependencies across frames and mitigates artifacts caused by local disturbances. Finally, we concatenate $x_T^0$ and $x_T^1$ along the frame dimension and feed them into the denoising process. Leveraging the model's forward denoising capability, we generate the corresponding next-frame image $x^1$. Since $x_T^1$ is sufficiently separated from the watermark-related distribution of $x_T^0$, the resulting image $x^1$ naturally removes the embedded watermark, thus achieving semantic-level watermark removal.

## 3.2 Watermark Removal: Optical Flow Transformation

To enable effective watermark removal, we propose an optimization algorithm over the optical flow matrix that maximizes the perceptual distance between the noise representations of consecutive frames in the latent space. This strategy drives the noise distribution away from watermark-related features, thereby decoupling watermark traces from the latent representation. Specifically, starting from the initial noise representation $x_T^0$, obtained via DDIM inversion of a watermarked image, we search for a two-dimensional optical flow matrix $f \in \mathbb{R}^{H \times W \times 2}$ within a constrained motion space $\mathcal{S}_\delta$. Applying this flow to $x_T^0$ yields a candidate next-frame noise representation $x_T^1$. The optimization objective is defined as:

$$f^* = \arg\max_{f \in \mathcal{S}_\delta} \ell_d \left( x_T^0, \mathcal{W}(x_T^0; f) \right), \tag{1}$$

where $\mathcal{W}(x_T^0; f) = x_T^1$ denotes the result of backward warping $x_T^0$ using flow $f$, and $\ell_d(\cdot)$ is a perceptual distance metric in latent space (e.g., $\ell_1$).

The search space $\mathcal{S}_\delta$ defines a bounded set of admissible flow perturbations, ensuring that the generated next-frame noise $x_T^1$ remains within a plausible local motion range relative to $x_T^0$. Formally, we define $\mathcal{S}_\delta$ as:

$$\mathcal{S}_\delta = \left\{ f \in \mathbb{R}^{H \times W \times 2} \mid |f_{i,j,k}| \leq \delta, \, \forall (i,j,k) \in [1, H] \times [1, W] \times [1, 2] \right\}, \tag{2}$$

where $f_{i,j,k}$ denotes the $k$-th component of the flow vector at spatial location $(i, j)$, and $\delta > 0$ is a predefined motion bound that constrains the maximum displacement per pixel along each spatial axis. This constraint ensures that the generated $x_T^1$ remains within the local motion subspace of $x_T^0$, preserving temporal coherence through spatial continuity. Simultaneously, $x_T^1$ exhibits a statistically distinct distribution from $x_T^0$, disrupting the latent consistency typically exploited by watermarking mechanisms. By integrating this adversarial flow search into the generation pipeline, we enable watermark removal in a black-box setting, without requiring any prior knowledge of the watermarking algorithm or the watermark carrier.

Motion in video sequences typically comprises two components: global transformations induced by camera motion and local variations caused by object motion within the scene. Since watermarked images may lack prominent dynamic objects, we focus on modeling camera-induced motion to guide semantic transformations. To this end, we construct flow-based transformation matrices that simulate

various camera motions by spatially warping the initial noise representation. We design three basic yet representative types of camera motion to evaluate the generality and effectiveness of NFPA under different motion conditions:

- **Horizontal motion (x-axis):** simulates lateral movement of the camera (e.g., from left to right).
- **Vertical motion (y-axis):** simulates vertical displacement of the camera (e.g., from top to bottom).
- **Combined horizontal and vertical motion (xy-axis):** simulates motion along both axes, where the camera can be moved in any direction within the image plane. The resulting flow matrix combines both horizontal and vertical components to form a more complex motion pattern.

It is important to note that NFPA is agnostic to specific camera motion patterns. Our attack does not rely on any fixed trajectory and can be extended to more complex or naturalistic motion types, such as camera zoom or rotation around an object.

### 3.3 Semantic Preservation: Frame-Attention

To ensure semantic and visual consistency between the next-frame prediction $x^1$ and the initial frame $x^0$, we introduce a modified frame-attention mechanism specifically designed for this task. We adapt the self-attention mechanism in the UNet backbone (i.e., Denoiser) into a frame-attention mechanism without modifying any model parameters. This mechanism explicitly conditions the generation of the next frame on the first frame in the sequence, thereby preserving semantic information such as object identity, spatial layout, and appearance across frames, despite the perturbations introduced for watermark removal.

In the original self-attention formulation, the input feature map $x \in \mathbb{R}^{h \times w \times c}$ is linearly projected into queries $Q$, keys $K$, and values $V$, and the attention output is computed as:

$$\text{Self-Attention}(Q, K, V) = \text{Softmax}\left(\frac{QK^\top}{\sqrt{n}}\right) V, \tag{3}$$

where $n$ denotes the embedding dimension of the features. In our frame-attention setting, we consider a sequence of two frames $x^{0:1} = [x^0, x^1] \in \mathbb{R}^{2 \times h \times w \times c}$, where $x^0$ serves as the reference frame. For each frame $x^k$, where $k \in \{0, 1\}$, we compute attention using the query $Q^k$ from frame $k$ and the key-value pairs $(K^0, V^0)$ from the reference frame $x^0$:

$$\text{Frame-Attention}(Q^k, K^0, V^0) = \text{Softmax}\left(\frac{Q^k (K^0)^\top}{\sqrt{n}}\right) V^0. \tag{4}$$

Importantly, the reference frame $x^0$ retains the standard self-attention mechanism to preserve its internal feature consistency and to provide a stable semantic anchor during each denoising timestep. Meanwhile, the next frame $x^1$ performs cross-attention with respect to $x^0$, explicitly inheriting semantic structures such as object arrangement and scene appearance from the reference frame.

This asymmetric attention formulation enforces a one-sided semantic dependency: $x^1$ is conditioned on $x^0$. As a result, the generated frame $x^1$ remains consistent with the original image in terms of semantics, while maintaining sufficient flexibility in the latent noise space to facilitate watermark removal. We find this design essential for balancing the competing objectives of watermark removal and semantic preservation. It maintains coherent object boundaries and improves perceptual fidelity. In practice, this frame-conditioning strategy enables NFPA to generate realistic and temporally coherent frames that are perceptually indistinguishable from the original watermarked images.

## 4 Evaluation and Analysis

### 4.1 Evaluation Setup

**Model and Dataset.** We use Stable Diffusion-v2.1-base (SD-v2.1) [2] as the default image generation model. SD-v2.1 is a widely adopted open-source generative model capable of producing high-fidelity images. Based on SD-v2.1 and the image-text descriptions from the MS-COCO-2017 [24] validation set, we generate AI-created images without watermarks to serve as the original images. We use 50 inference steps to generate all images and set a random seed for each image to eliminate the influence

of stochastic variation. On this basis, we apply various image watermarking methods to produce the corresponding watermarked images. All experiments are conducted on a machine equipped with an Nvidia GeForce RTX 4090 GPU.

**Proposed Attack Setup.** We construct a next-frame prediction framework based on SD-v2.1 by default and set the maximum search range $\delta$ of the optical flow matrix to 40, limiting the motion range of the next-frame image to between -40 and 40 pixels. By adjusting $\delta$ allows us to trade-off Quality-detectability, see Appendix D for details. For DDIM inversion, we set the number of inference steps to 10 by default to improve attack efficiency and use an empty prompt during the inversion process, as the prompt for the watermarked image is unknown during the attack. In the subsequent experimental section, we perform ablation studies to examine the impact of hyperparameters such as the base model, frame-attention, and inference steps.

**Watermark Baselines.** We consider four post-processing watermark methods, including Dwt-Dct [9], RivaGAN [46], StegaStamp [41], and SSL Watermarking [12], as well as four in-processing watermark methods, including Tree-Ring [43], RingID [8], StableSignature [13], and GaussianShading [44]. These methods cover a range of techniques, from traditional pixel-level watermarking to the latest semantic-level watermarking. The watermark embedding process strictly follows the default configurations provided in the official implementations of each method, with detailed information provided in Appendix A. In the watermark detection phase, we follow prior work [48, 43], and set the decision threshold to reject the null hypothesis at a significance level of $p < 0.01$. The null hypothesis $H_0$ assumes that the image does not contain an embedded watermark. Formally, this hypothesis is defined as: $H_0 : \sum_{i=\tau+1}^{n} \binom{n}{i}(0.5)^n < 0.01$, where $n$ denotes the total number of embedded watermark bits, and $\tau$ is the minimum number of correctly extracted bits required to reject $H_0$. For example, when embedding a $n = 32$ bit watermark, if at least 23 bits are correctly extracted, $H_0$ can be rejected, indicating that the image contains a watermark.

**Attack Baselines.** To comprehensively evaluate the performance of our method in removing image watermarks, we consider seven distortion attack baselines: Rotation, JPEG Compression, Cropping & Scaling, Gaussian Blur, Gaussian Noise, Color Jitter, and Translation. We also consider four regeneration attack baselines: Diffusion-Attack (DA) [48, 37], VAE-Attack (VA) [48], CtrlRegen+ [25], and a comparison baseline that uses Stable Video Diffusion (SVD) [5] to predict the next frame. In addition, we consider two adversarial attack baselines: model substitution adversarial attack (MSAA) [37] and IRA [28]. Because there is a trade-off between watermark removal effectiveness and image quality, we adjust the attack parameters to faithfully reflect the performance of each baseline, ensuring they produce similar image quality for a fair comparison. Under these settings, the attack with the lowest watermark verification accuracy can reasonably be considered the most effective. See Appendix B for detailed attack parameters.

**Evaluation Metrics.** Following prior work [43, 48, 15], we adopt the true positive rate at a false positive rate of $1\%$ (TPR@1%FPR) as the metric for evaluating watermark robustness. This setting aligns with the null hypothesis defined in watermarking baselines, and it quantifies the ability of the watermark detector to reliably identify watermarked images while maintaining a low false positive rate. To assess the fidelity of attacked images relative to their originals, we use the Frechet Inception Distance (FID) [17] to measure image quality and the CLIP score [33, 7] to evaluate the semantic consistency between the image and its associated prompt. For TPR@1%FPR, we apply all attack baselines to 1,000 watermarked images to compute the metric. To evaluate the FID and CLIP scores, we compute them over 1,000 attacked watermarked images and 1,000 corresponding real image-text pairs from the MS-COCO-2017 validation set. Since our attacks operate at the semantic level, pixel-level metrics such as PSNR and SSIM are not applicable, and thus are excluded from this study.

## 4.2 Evaluation Results

We conducted experiments on eight image watermarking schemes and validated the performance of NFPA by comparing it against thirteen watermark removal attacks. Our extensive experiments aim to answer the following research questions (RQs):

- [RQ1] How effective is NFPA in removing image watermarks?
- [RQ2] How well does NFPA preserve image quality?
- [RQ3] How efficient is NFPA in executing attacks?
- [RQ4] How do different modules affect the performance of NFPA?

Table 1: [RQ1] Watermark removal performance of the removal attack across eight image watermarking methods, evaluated using TPR@1%FPR. Lower values indicate more effective removal. **Bolded** values denote the best performance; *underlined italicized* values indicate the second best.

| Attack | DwtDct | RivaGAN | SSL | StegaStamp | TreeRing | StableSignature | RingID | GaussianShading | Avg. |
|---|---|---|---|---|---|---|---|---|---|
| None | 0.79 | 1.00 | 1.00 | 1.00 | 1.00 | 1.00 | 1.00 | 1.00 | 0.97 |
| JPEG Compression | *0.01* | 0.59 | 0.08 | 1.00 | 0.97 | 0.79 | 1.00 | 0.99 | 0.68 |
| Cropping & Scaling | *0.01* | 0.96 | 0.90 | **0.00** | *0.05* | 0.99 | **0.01** | **0.00** | 0.36 |
| Gaussian Blur | 0.14 | 1.00 | 1.00 | 1.00 | 1.00 | 0.66 | 1.00 | 1.00 | 0.85 |
| Gaussian Noise | **0.00** | 0.86 | 0.02 | 1.00 | 0.99 | 0.41 | 1.00 | 0.99 | 0.66 |
| Color Jitter | 0.16 | 0.86 | 0.62 | 0.99 | 0.97 | 0.96 | 0.99 | *0.98* | 0.82 |
| Rotation | *0.01* | 0.99 | 1.00 | 0.45 | 0.21 | 0.98 | 1.00 | **0.00** | 0.58 |
| Translation | 0.03 | 1.00 | 1.00 | 0.17 | 0.38 | 1.00 | 0.31 | *0.01* | 0.49 |
| VA | 0.02 | 0.73 | 0.34 | 1.00 | 1.00 | 0.95 | 1.00 | 1.00 | 0.75 |
| DA | *0.01* | *0.05* | *0.01* | 0.72 | 0.92 | **0.00** | 0.99 | 0.99 | 0.46 |
| CtrlRegen+ | 0.01 | **0.02** | 0.03 | 0.36 | 0.73 | **0.00** | 0.96 | 1.00 | 0.39 |
| SVD | 0.10 | 0.53 | 0.51 | 1.00 | 0.91 | *0.10* | 0.99 | 0.76 | 0.61 |
| IRA | 0.02 | **0.02** | **0.00** | 0.15 | **0.04** | **0.00** | 0.14 | **0.00** | *0.05* |
| MSAA | - | - | - | 1.00 | 0.07 | - | - | - | 0.53 |
| NFPA-x | *0.01* | 0.15 | 0.16 | 0.20 | 0.25 | **0.00** | 0.16 | **0.00** | 0.12 |
| NFPA-y | *0.01* | 0.13 | 0.16 | 0.14 | 0.22 | **0.00** | 0.10 | **0.00** | 0.10 |
| NFPA-xy (Ours) | *0.01* | 0.13 | 0.09 | *0.02* | 0.07 | **0.00** | *0.02* | **0.00** | **0.04** |

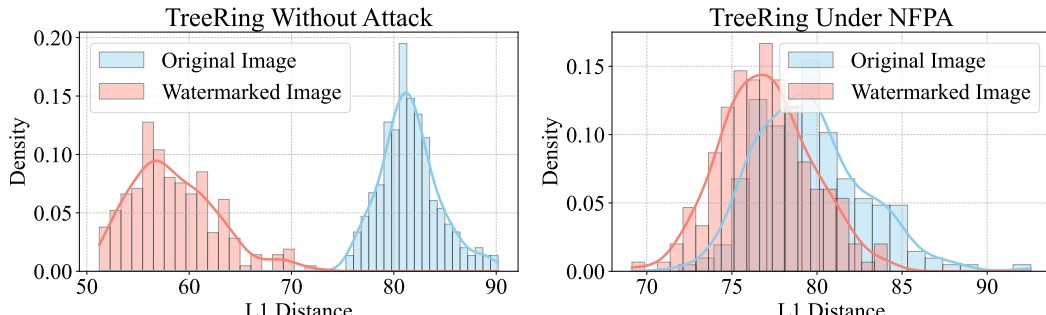

Figure 3: [RQ1] Histogram distributions of TreeRing before and after attack. Images with smaller $\ell_1$ distances are more likely to contain watermarks.

**RQ1: Analysis of Watermark Removal Effectiveness.** Table 1 reports the detection accuracy (TPR@1%FPR) of eight representative image watermarking methods under various attack strategies. We include MSAA only for TreeRing and StegaStamp due to their high computational cost, as it requires training watermark-specific surrogate detectors. We use the official pretrained models released by the authors for these two watermarking methods.

We summarize four key observations from the results: *First*, traditional distortion attacks (e.g., JPEG compression, blurring, noise) generally fail to remove watermarks effectively. Most watermarking methods still exhibit high detection rates after such perturbations, indicating strong robustness to low-level image degradations. *Second*, regeneration attacks such as diffusion attack (DA) are more effective, particularly for pixel-level watermarking methods like DwtDct and Stable Signature. For these methods, the TPR drops substantially demonstrating that latent-space regeneration can disrupt low-level watermark features. However, DA remains less effective against semantic-level watermarks such as TreeRing and RingID, which are embedded through high-level features. *Third*, adversarial attacks such as MSAA show greater potential in removing semantic-level watermarks. Nevertheless, they incur high attack costs, including training dedicated agent models or running iterative optimization loops, which limits their scalability and practicality. *Finally*, NFPA achieves consistently superior performance across all evaluated watermarking methods. Regardless of the motion strategy employed (e.g., x-axis, y-axis, or combined xy-axis), NFPA significantly reduces detection accuracy for both pixel-level and semantic-level watermarks. It achieves the lowest average TPR@1%FPR (0.04) across all attacks. As illustrated in Figure 3, NFPA substantially increases the $\ell_1$ distance between the attacked images and the watermark patterns of TreeRing, rendering them indistinguishable from unwatermarked images. These results validate our approach that decoupling watermark signals in the latent space (formulated through the optimization in Equation 1) is an effective strategy for watermark removal.

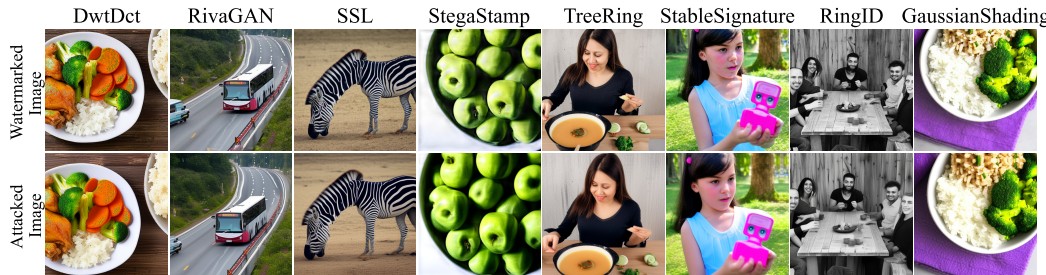

Figure 4: [RQ2] Attacked image examples of NFPA for eight image watermarking schemes.

It is worth noting that although IRA demonstrates performance comparable to ours, it is inherently an adversarial attack whose effectiveness heavily depends on the architectural similarity between the target and surrogate models. As reported in the original paper, when the target and surrogate models differ (e.g., SD2.1 vs. SDXL), IRA fails to reduce the TreeRing watermark detection rate below 0.33 even after 100 optimization steps. In our implementation, both the target and surrogate models use SD2.1, corresponding to IRA's white-box setting and effectively representing its upper-bound performance. In contrast, NFPA operates as a stable black-box attack that makes no assumptions about the target model's architecture. This design enables our method to achieve state-of-the-art performance in terms of watermark removal effectiveness and practicality.

**RQ2: Analysis of Image Quality Preservation.** To evaluate the impact of removal attacks on visual quality, we report the average FID and CLIP score across eight watermarking methods in Table 2, with full results provided in Appendix C. Notably, we calibrate the parameters of all evaluated attacks to ensure comparable image quality, thereby enabling a fair comparison of watermark removal effectiveness. NFPA achieves image quality on par with the original images, demonstrating its ability to preserve visual fidelity during the watermark removal process. Crucially, under similar quality conditions, NFPA outperforms all other baselines in watermark removal performance. These results highlight the trade-off achieved by NFPA between effective watermark removal and high perceptual quality. Examples in Figure 4 further support these findings, showing that the subtle semantic modifications introduced via next-frame prediction have negligible perceptual impact. Additional image examples for baseline attacks are provided in Appendix C.

Furthermore, we control the attack intensity by adjusting the corresponding parameters and present the resulting quality–detectability trade-off in Appendix D. NFPA consistently achieves the optimal Pareto frontier across all evaluated scenarios.

Table 2: [RQ2] Average FID and CLIP scores of watermarked images under attack.

| Attack | FID↓ | CLIP↑ |
|---|---|---|
| None | 66.57 | 0.33 |
| JPEG | 73.42 | 0.33 |
| Crop | 69.33 | 0.32 |
| Blur | 73.99 | 0.33 |
| Noise | 69.98 | 0.32 |
| Color Jitter | 70.84 | 0.32 |
| Rotation | 73.63 | 0.32 |
| Translation | 68.15 | 0.31 |
| VA | 70.92 | 0.33 |
| DA | 74.89 | 0.32 |
| CtrlRegen+ | 66.60 | 0.32 |
| SVD | 67.85 | 0.32 |
| IRA | 66.60 | 0.32 |
| MSAA | 73.34 | 0.32 |
| NFPA-x | 69.40 | 0.32 |
| NFPA-y | 69.39 | 0.32 |
| NFPA-xy | 69.48 | 0.32 |

**RQ3: Analysis of Attack Efficiency.** Distortion attacks involve simple transformations and incur only millisecond-level overhead. In contrast, regeneration and adversarial attacks are more effective but substantially more costly, e.g., IRA takes on average approximately five minutes per image, as shown in Table 3. In this context, NFPA achieves a favorable balance between attack effectiveness and execution efficiency. By leveraging our novel next-frame prediction framework, NFPA substantially reduces the computational cost of video generation, lowering the average removal time for a single watermarked image to 1.2 seconds. This design enables NFPA to maintain strong attack performance while ensuring practical runtime efficiency, outperforming existing baselines in terms of overall attack utility.

Table 3: [RQ3] Time cost of different attacks.

| Attack | Time (s/image) |
|---|---|
| VA | $0.01_{3.15 \times 10^{-3}}$ |
| DA | $0.30_{0.01}$ |
| CtrlRegen+ | $2.41_{0.48}$ |
| SVD | $79.17_{3.62}$ |
| IRA | $323.87_{0.65}$ |
| MSAA | $2.48_{0.25}$ |
| NFPA-xy | $1.27_{0.02}$ |

**RQ4: Ablation Study of Different Components.** We perform a systematic ablation study to quantify how the choice of base model, the inclusion of the frame-attention mechanism, and the number of inference denoising steps affect NFPA 's watermark removal performance and perceptual quality. All

Figure 5: [RQ4] Ablation study results for base models.

| Base Model | TreeRing | | | Stable Signature | | |
|---|---|---|---|---|---|---|
| | T@1%F ↓ | FID ↓ | CLIP Score ↑ | T@1%F ↓ | FID ↓ | CLIP Score ↑ |
| SD-v1.4 | 0.05 | 69.38 | 0.33 | 0.00 | 69.07 | 0.33 |
| SD-v1.5 | 0.04 | 69.47 | 0.33 | 0.00 | 69.04 | 0.33 |
| SD-v2.0 | 0.07 | 69.35 | 0.33 | 0.00 | 68.83 | 0.33 |
| SD-v2.1 | 0.07 | 69.18 | 0.33 | 0.00 | 68.87 | 0.33 |

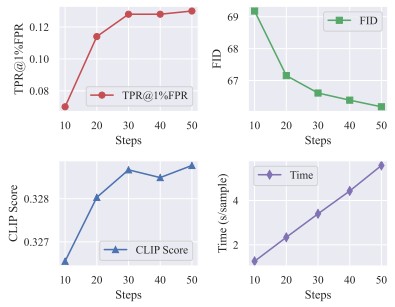

| Watermarked Image | NFPA (Frame-Attention) | NFPA (Self-Attention) | Watermarked Image | NFPA (Frame-Attention) | NFPA (Self-Attention) |

Figure 6: [RQ4] Effect of frame-attention on image quality.

Figure 7: [RQ4] Performance across inference steps for TreeRing watermark.

ablations use two representative watermarking schemes, TreeRing and Stable Signature, to ensure conclusions generalize across different watermark types.

First, Table Figure 5 presents results obtained with several diffusion backbones, ranging from SD-v1.4 to SD-v2.1. As the capacity of the base model increases, we observe a modest improvement in perceptual image quality. Importantly, NFPA 's ability to remove watermarks remains consistently strong across all tested model versions. This stability demonstrates that NFPA is largely model agnostic and that its removal capability does not rely on a particular backbone architecture.

Second, Figure Figure 6 compares outputs produced with and without the frame-attention module. Removing this module causes a clear and substantial drop in visual fidelity and inter-frame semantic coherence. These results confirm that the frame-attention mechanism is essential for preserving semantic consistency across frames while allowing effective watermark removal.

Third, Figure Figure 7 analyzes the influence of the number of denoising steps used during inference. Increasing the number of steps yields finer denoising and therefore better pixel-level reconstruction and perceptual quality. At the same time, more steps can slightly reduce attack strength because the reconstruction becomes closer to the original image, which can preserve some watermark evidence at the semantic level. To strike a practical balance between removal effectiveness and visual quality, we adopt 10 denoising steps as the default setting. We also study the role of the maximum optical flow search range, denoted by $\delta$, in Appendix D. When $\delta$ is set to zero, no motion is introduced and NFPA reduces to a latent regeneration attack that performs poorly. As $\delta$ grows, the method can explore a larger motion and feature space to better decouple watermark signals from content, but larger values of $\delta$ may introduce more perceptual distortion. This parameter therefore provides a controllable trade-off between detection evasion and image quality.

## 5 Conclusion

In this work, we introduce NFPA, the first semantic-level image watermark removal method, which reveals the vulnerabilities of state-of-the-art watermarking techniques. Leveraging our next-frame prediction model, NFPA effectively addresses the dual challenges of watermark removal and semantic preservation. Extensive experimental results demonstrate that NFPA achieves SOTA watermark removal performance, while striking an optimal balance between image quality and attack efficiency. Our findings highlight the inherent weaknesses of current watermarking approaches, underscoring the urgent need for stronger defense mechanisms in AI-generated image watermarking.

## Acknowledgement

We are thankful to the shepherd and reviewers for their careful assessment and valuable suggestions, which have helped us improve this paper. This work was supported in part by the National Natural Science Foundation of China (62472096, 62172104, 62172105, 62102093, 62102091, 62302101, 62202106). Min Yang is a faculty of the Shanghai Institute of Intelligent Electronics & Systems and Engineering Research Center of Cyber Security Auditing and Monitoring, Ministry of Education, China.

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

# A  Watermark Baselines Descriptions

- **DwtDct** [9]: DwtDct is a traditional watermark scheme that implants the watermark in the frequency domain by using a combination of Discrete Wavelet Transform (DWT) and Discrete Cosine Transform (DCT). For implementation, we utilize a popular Python library named *invisible-watermark* [2], and we set the watermark length to 30 bits by default.

- **RivaGAN** [46]: RivaGAN is an encoder-decoder based watermark scheme originally used for video watermarking, and can also be adapted for robust image watermarking. This watermark incorporates a custom attention-based mechanism for embedding arbitrary data. Additionally, it employs two independent adversarial networks to maintain video quality and enhance watermark robustness through adversarial optimization. For implementation, we also leverage *invisible-watermark* [2] and we set the watermark length to 32 bits by default.

- **StegaStamp** [41]: StegaStamp also uses a trained encoder neural network to embed the watermark information, and then leverages a trained decoder neural network to detect the embedded watermark. However, it introduces more human-visible artifacts, compromising the image quality. For implementation, we use the code and the pretrained model published by the original authors [3] and we set the watermark length to 100 bits by default.

- **SSL** [12]: SSL embeds watermarks in self-supervised-latent spaces by shifting the feature of the image to a selected region. For implementation, we use the code and the pretrained model published by the original authors [4] and we set the watermark length to 30 bits by default.

- **StableSignature** [13]: StableSignature is an in-process watermark mechanism that embeds the watermark during the image generation process. It leverages the watermark decoder from HiDDeN [49], and then fine-tunes the VAE decoder of the latent-diffusion model (LDM) to ensure the watermark information can be decoded from the generated images. For implementation, we use the code and the pretrained model published by the original authors [5] and we set the watermark length to 48 bits by default.

- **TreeRing** [43]: TreeRing also belongs to the in-process watermark mechanism. Differing from StableSignature, TreeRing modifies the initial seed of the diffusion model by embedding a predefined ring pattern. In the verification stage, the specific pattern can be detected by using the DDIM inversion process on the watermark images. For implementation, we use the code published by the original authors [6].

- **RingID** [8]: Based on TreeRing, RingID also embeds the watermark information by modifying the initial seed of the diffusion model. It identifies the limitations in TreeRing's design and introduces a series of approaches for enhanced distinguishability and robustness. For implementation, we use the code published by the original authors [7].

- **GaussianShading** [44]: Based on TreeRing, GaussianShading develops an initial seed modification watermark that doesn't deteriorate the generated image quality by watermark randomization and distribution preserving sampling. For implementation, we use the code published by the original authors [8] and we set the watermark length to 256 bits by default.

Table 4: Watermark length of different watermark schemes. (Note that TreeRing and RingID watermarks are not binary; thus, this metric is not applicable.)

| Watermarking | DwtDct | RivaGAN | StegaStamp | SSL | StableSignature | GaussianShading |
|---|---|---|---|---|---|---|
| Length (# bits) | 30 | 32 | 100 | 30 | 48 | 256 |

---

[2] https://github.com/ShieldMnt/invisible-watermark
[3] https://github.com/tancik/StegaStamp
[4] https://github.com/facebookresearch/ssl_watermarking
[5] https://github.com/facebookresearch/stable_signature
[6] https://github.com/YuxinWenRick/tree-ring-watermark
[7] https://github.com/showlab/RingID
[8] https://github.com/bsmhmmlf/Gaussian-Shading

# B    Attack Baselines Descriptions

- **JPEG Compression**: It is a widely used image compression method, characterized by a quality factor parameter that controls the degree of compression. A lower quality factor results in greater loss of fine image details and increases the likelihood of watermark degradation. In Section 4, the quality factor is set to 20. In Appendix D, the quality factors are set to 10, 20, 30, 40, 50, and 60.

- **Cropping & Scaling**: Crop is a common image processing operation. It maintains portions of an image based on the selected crop ratio. A higher crop ratio means more portions remain. Excessive cropping can result in loss of critical image content and may affect the integrity of the embedded watermark. In our experiments, we employ a center crop strategy followed by resizing the cropped image to its original size. In Section 4, the crop ratio is set to 0.7. In Appendix D, the crop ratios are 0.6, 0.65, 0.7, 0.75, 0.8, and 0.85.

- **Gaussian Blur**: It involves convolving the watermark image with a kernel, such as a Gaussian kernel, to make the watermark less detectable. The kernel size determines the degree of distortion applied to the watermark image, and a larger kernel size results in stronger attack performance. In Section 4, a Gaussian kernel with a size of 15 is used. In Appendix D, kernel sizes of 5, 7, 9, 11, 13, and 15 are employed.

- **Gaussian Noise**: It introduces Gaussian noise, to each pixel of the watermark images to distort the watermark information. For Gaussian noise, the variance determines the strength of the added noise. A higher variance causes the watermark image to lose more information. In Section 4, the variance is set to 30. In Appendix D, variances of 10, 15, 20, 25, 30, 35, and 40 are used.

- **Color Jitter**: It modifies the brightness of the watermark images by scaling all the pixels in the watermark images. Larger brightness changes make the watermark harder to detect. In Section 4, the brightness factor is set to 4. In Appendix D, brightness factors of 1, 2, 3, 4, 5, and 6 are applied.

- **Rotation**: It is a common geometric transformation that alters the orientation of an image by a specified angle. Larger rotation angles can cause misalignment of the embedded information and increase the risk of watermark distortion. In Section 4, the rotation angle is set to 10 degrees. In Appendix D, rotation angles of 5, 6, 7, 8, 9, and 10 degrees are considered.

- **VAE-Attack (VA)** [48]: VA represents a type of regeneration attack, and can remove the watermark during the regeneration process. The watermark image is first mapped to the latent space by the VAE encoder and then mapped to the pixel space by the VAE decoder. Both the encoder and decoder are parameterized with neural networks. Specifically, we utilize the VAE-Bmshj2018 [9] to perform the VA. The compression factor controls the attack strength of VA, with lower values corresponding to stronger attacks. In Section 4, the compression factor is set to 5. In Appendix D, compression factors of 3, 4, 5, 6, 7, and 8 are used.

- **Diffusion-Attack (DA)** [48]: DA utilizes a diffusion model to first add noise to the watermark image to eliminate the watermark, and then uses the reverse process to reconstruct the image. Increasing the number of noise steps introduces more noise, thus resulting in better attack performance. In our experiments, we use the Stable Diffusion-v2.1-base [2] to perform the DA. In Section 4, the number of noise steps is set to 100. In Appendix D, noise steps of 70, 80, 90, 100, 110, and 120 are evaluated.

- **Model Substitute Adversarial Attack (MSAA)** [37]: MSAA involves training a substitute classifier and conducting projected gradient descent (PGD) attacks on it to deceive black-box watermark detectors. The perturbation budget $\epsilon$ in the PGD attack controls the attack strength of MSAA, and a larger perturbation budget induces a stronger attack performance. In Section 4, the perturbation budget is set to 8. In Appendix D, perturbation budgets of 5, 6, 7, 8, 9, and 10 are used.

- **Imprint-Removal Attack (IRA)** [28]: IRA is an attack designed to remove semantic watermarks using a black-box proxy model. The attack first maps the watermarked image

---

[9] https://github.com/InterDigitalInc/CompressAI

Table 5: [RQ2] FID of watermarked images under attack. Lower values indicate better semantic consistency.

| Attack | DwtDct | RivaGAN | SSL | StegaStamp | TreeRing | StableSignature | RingID | GaussianShading | Avg. |
|---|---|---|---|---|---|---|---|---|---|
| None | 65.22 | 65.60 | 64.89 | 67.79 | 67.31 | 66.48 | 68.32 | 66.99 | 66.57 |
| JPEG | 72.09 | 72.31 | 73.16 | 79.56 | 71.99 | 72.41 | 73.13 | 72.75 | 73.42 |
| Crop | 68.22 | 67.88 | 67.09 | 71.19 | 69.87 | 68.25 | 71.00 | 71.16 | 69.33 |
| Blur | 71.78 | 70.52 | 70.07 | 82.29 | 74.16 | 73.10 | 75.40 | 74.64 | 73.99 |
| Noise | 69.54 | 68.89 | 70.01 | 73.69 | 68.26 | 69.12 | 70.50 | 69.84 | 69.98 |
| Color Jitter | 70.60 | 69.44 | 71.18 | 72.41 | 70.00 | 69.09 | 71.23 | 72.77 | 70.84 |
| Rotation | 72.79 | 71.12 | 72.85 | 79.32 | 72.56 | 71.55 | 74.53 | 74.28 | 73.63 |
| VA | 69.60 | 69.97 | 69.84 | 74.42 | 70.66 | 70.53 | 71.66 | 70.68 | 70.92 |
| DA | 70.29 | 69.69 | 70.08 | 70.82 | 78.25 | 79.07 | 82.17 | 78.74 | 74.89 |
| IRA | 66.11 | 64.90 | 66.14 | 69.70 | 65.53 | 66.04 | 67.83 | 66.52 | 66.60 |
| CtrlRegen+ | 66.06 | 66.24 | 65.60 | 65.76 | 67.10 | 65.97 | 68.72 | 67.32 | 66.60 |
| MSAA | - | - | - | 74.67 | 72.01 | - | - | - | 73.34 |
| SVD | 66.11 | 66.80 | 66.10 | 69.29 | 68.55 | 67.45 | 69.58 | 68.91 | 67.85 |
| Translation | 66.51 | 66.48 | 65.82 | 71.65 | 68.43 | 67.22 | 69.77 | 69.30 | 68.15 |
| NFPA-x | 67.74 | 68.54 | 67.89 | 70.00 | 69.65 | 69.85 | 71.74 | 69.77 | 69.40 |
| NFPA-y | 68.08 | 68.73 | 67.95 | 69.84 | 69.93 | 69.17 | 71.40 | 69.99 | 69.39 |
| NFPA-xy (Ours) | 67.52 | 68.60 | 71.47 | 69.56 | 69.18 | 68.87 | 71.09 | 69.54 | 69.48 |

to the proxy model's latent space and inverts it to estimate the latent noise vector. It then performs gradient descent to find a perturbation for the latent image, optimizing a loss function that encourages the new inverted latent noise to be dissimilar to the original one (e.g., by targeting its negation). The strength of the attack is controlled by the number of optimization steps. More steps generally improve watermark removal but can increase image distortion. In the experiments, we set the number of optimization steps to 50 by default.

- **CtrlRegen+** [25]: CtrlRegen+ is an adjustable watermark removal method that uses a controllable regeneration process. The method first encodes the watermarked image into its latent representation and adds a specified number of noise steps to create a noisy latent. It then uses a controllable diffusion model, guided by semantic and spatial features extracted from the original watermarked image, to denoise this latent and reconstruct the image. The number of noise steps controls the attack strength; more steps lead to more thorough watermark removal, particularly for high-perturbation watermarks, while the control networks maintain high image quality and consistency. In the experiment, we set the number of noise steps to 500 following the source code.

- **SVD** [5]: It uses Stable Video Diffusion (SVD), a trained latent diffusion model for generating short video clips from a single conditioning image. We use the watermarked image and an empty text prompt as input conditions to generate the video. To align with our method's setup, we by default use the frame at generated video index 1 (i.e., the next frame) as the attacked image.

- **Translation**: Translation is a common geometric transformation that shifts an image horizontally or vertically by a specified number of pixels. This spatial displacement interferes with the detection of embedded watermarks, especially when the detection process depends on position. Larger translation distances cause more severe displacement and increase the risk of watermark distortion. To align with our method's setup, we set the translation distance to 40 pixels in both the horizontal and vertical directions.

## C   Analysis of Image Quality Preservation

To comprehensively evaluate the impact of removal attacks on image quality, we report detailed results for all eight watermarking schemes in terms of FID and CLIP score. As shown in Table 5 and Table 6, NFPA consistently achieves FID and CLIP scores comparable to those of the original images, indicating negligible degradation in visual fidelity. These results confirm that NFPA effectively preserves perceptual quality while successfully removing watermarks. Figure 8 presents image examples of attacked images produced by each method across all watermarking schemes. Based on our next-frame prediction frame, NFPA introduces only minor semantic modifications, resulting in images that remain visually indistinguishable from their watermarked counterparts. These qualitative results further support the effectiveness of our approach in maintaining high image fidelity across diverse watermarking scenarios.

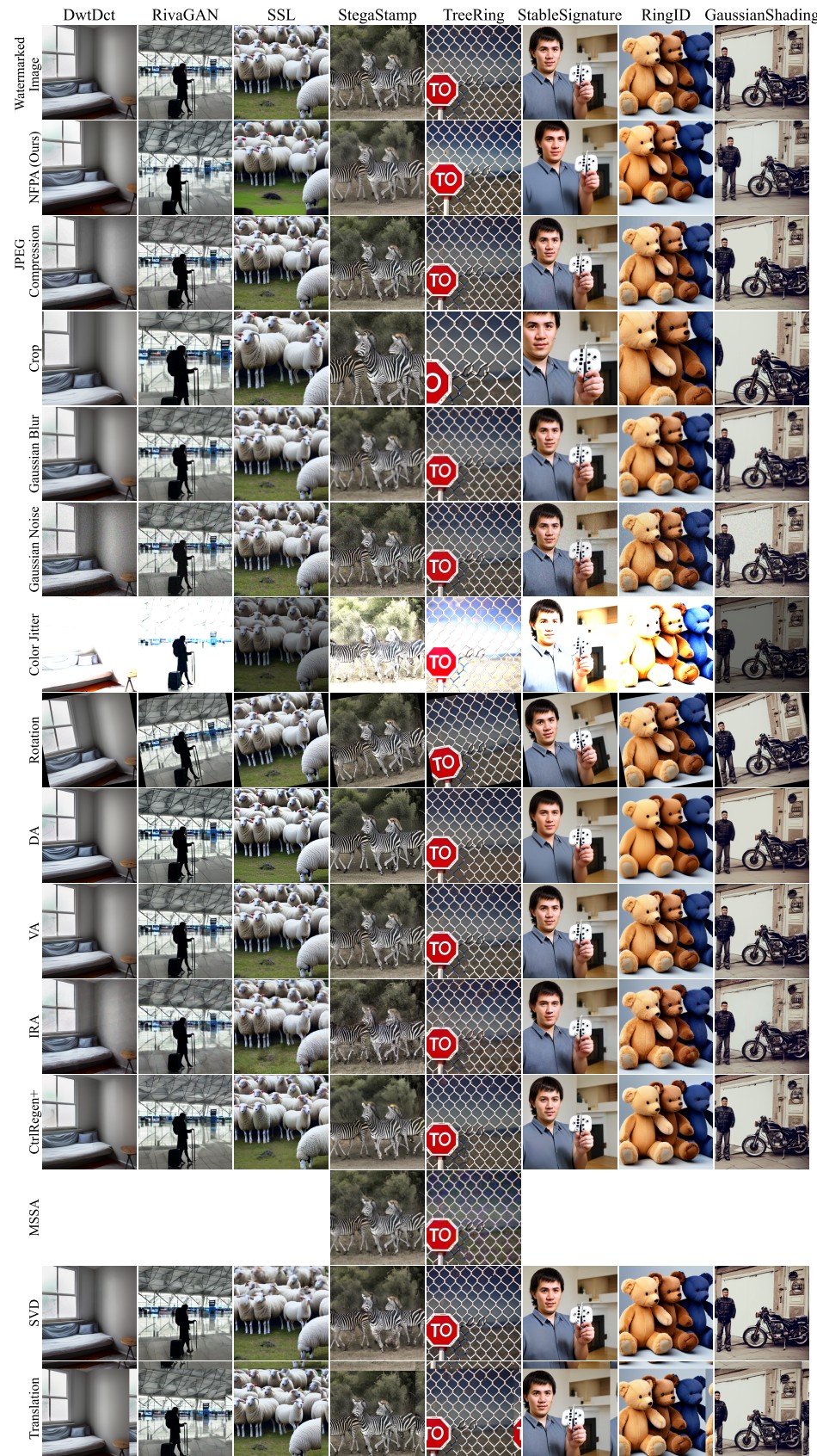

Figure 8: [RQ2] Attacked image examples for eight image watermarking schemes.

Table 6: [RQ2] CLIP score of watermarked images under attack. Higher values indicate better image quality.

| Attack | DwtDct | RivaGAN | SSL | StegaStamp | TreeRing | StableSignature | RingID | GaussianShading | Avg. |
|---|---|---|---|---|---|---|---|---|---|
| None | 0.32 | 0.33 | 0.33 | 0.32 | 0.33 | 0.33 | 0.33 | 0.33 | 0.33 |
| JPEG | 0.33 | 0.33 | 0.33 | 0.33 | 0.33 | 0.33 | 0.33 | 0.33 | 0.33 |
| Crop | 0.31 | 0.32 | 0.32 | 0.32 | 0.32 | 0.32 | 0.32 | 0.32 | 0.32 |
| Blur | 0.32 | 0.33 | 0.33 | 0.32 | 0.33 | 0.33 | 0.33 | 0.33 | 0.33 |
| Noise | 0.32 | 0.32 | 0.32 | 0.31 | 0.32 | 0.32 | 0.32 | 0.32 | 0.32 |
| Color Jitter | 0.32 | 0.32 | 0.32 | 0.31 | 0.32 | 0.32 | 0.32 | 0.32 | 0.32 |
| Rotation | 0.32 | 0.32 | 0.32 | 0.32 | 0.33 | 0.33 | 0.32 | 0.33 | 0.32 |
| VA | 0.33 | 0.33 | 0.33 | 0.33 | 0.33 | 0.33 | 0.33 | 0.33 | 0.33 |
| DA | 0.32 | 0.33 | 0.33 | 0.32 | 0.32 | 0.32 | 0.32 | 0.32 | 0.32 |
| IRA | 0.31 | 0.32 | 0.32 | 0.31 | 0.32 | 0.32 | 0.32 | 0.32 | 0.32 |
| CtrlRegen+ | 0.31 | 0.32 | 0.32 | 0.32 | 0.32 | 0.32 | 0.32 | 0.32 | 0.32 |
| MSAA | - | - | - | 0.32 | 0.32 | | | - | 0.32 |
| SVD | 0.32 | 0.33 | 0.32 | 0.32 | 0.32 | 0.32 | 0.32 | 0.33 | 0.32 |
| Translation | 0.31 | 0.31 | 0.31 | 0.32 | 0.32 | 0.32 | 0.31 | 0.32 | 0.31 |
| NFPA-x | 0.32 | 0.32 | 0.31 | 0.32 | 0.33 | 0.33 | 0.32 | 0.33 | 0.32 |
| NFPA-y | 0.32 | 0.32 | 0.31 | 0.32 | 0.33 | 0.33 | 0.32 | 0.33 | 0.32 |
| NFPA-xy | 0.32 | 0.32 | 0.31 | 0.32 | 0.33 | 0.33 | 0.32 | 0.33 | 0.32 |

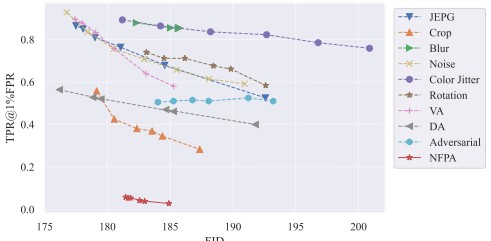

Figure 9: Trade-off between detectability and FID averaged over eight watermarking schemes tested against all attack methods.

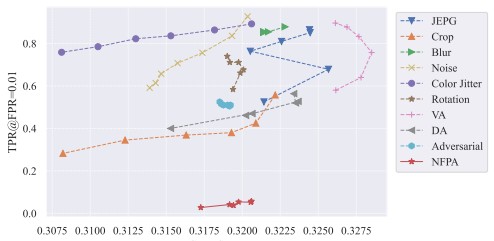

Figure 10: Trade-off between detectability and CLIP score averaged over eight watermarking schemes tested against all attack methods.

## D    Quality-Detectability Tradeoff

To further analyze the balance between watermark detectability and visual quality, we adjust the hyperparameters of each attack method (detailed in Appendix B) to control the attack intensity, thereby characterizing their performance in the quality–detectability trade-off. Figure 9 and Figure 10 illustrate the relationship between detectability and visual quality, as measured by FID and CLIP scores, respectively. Each curve represents a specific attack method, and the results are averaged over eight watermarking schemes to reflect the overall performance.

The results show that NFPA consistently lies on or near the optimal Pareto frontier across. Specifically, NFPA achieves low average detectability while maintaining high perceptual quality, demonstrating superior overall attack effectiveness. This indicates that our semantic-level watermark removal approach effectively balances attack strength and image fidelity. In contrast, distortion-based methods (e.g., JPEG, cropping, noise) generally exhibit better visual quality but limited ability to reduce detectability. Adversarial and regeneration-based attacks, while sometimes effective, fail to consistently suppress detectability across all watermark types, leading to suboptimal removal performance. In comparison, NFPA bridges this gap and consistently reduces watermark detectability across all evaluated schemes, significantly outperforming existing baselines in the quality–detectability trade-off.

## E    Limitations and Future Work

In this study, we take an important step toward understanding and revealing the vulnerabilities of existing image watermarking schemes. The proposed attack relies on a pretrained T2I diffusion model to predict the next frame image. While this strategy already demonstrates promising results, future work may explore customized or fine-tuned video generation models to further improve fidelity

and consistency. In addition, the quality of the generated images in our attack is affected by the accuracy of DDIM inversion. Notably, the inherent information loss during the inversion process may be mitigated by constructing more precise and reversible generative trajectories [18, 32, 42], which could further enhance the attack performance.

In formalizing NFPA, we introduce several basic camera motion patterns, such as planar translation. Experimental results show that even with such simple motion patterns, NFPA is already effective at removing SOTA image watermarks. Future research may explore more complex camera motion patterns to assess their potential and advantages in more challenging watermarking scenarios. In our evaluation, we have tried our best to cover influential watermarking schemes published in recent top-tier conferences. Future works may consider further validating our attack in more newly proposed image watermarking.

## F   Societal Impacts

This work proposes NFPA and reveals, for the first time, the potential vulnerabilities of several influential image watermarking schemes when facing semantic perturbations. Although such attack methods carry the risk of malicious exploitation, we emphasize that the core purpose of this paper is to promote a deeper understanding and open discussion of the limitations of current watermark defense mechanisms, thereby advancing the overall security in this field. Through an extensive evaluation of multiple mainstream image watermarking schemes, we demonstrate that these watermarks can be effectively removed in practical application scenarios, highlighting the urgent need to design more robust watermarking mechanisms. As the first attack framework that transforms the watermark removal problem into a video frame prediction task, NFPA provides a novel validation benchmark for developing the next generation of image watermarking techniques capable of resisting semantic-level perturbations. In general, we believe that this study contributes to the development of image watermarking techniques that will ultimately enhance the detectability and traceability of AI-generated images.

