# OpenReview forum: "The Future Unmarked: Watermark Removal in AI-Generated Images via Next-Frame Prediction"
_NeurIPS.cc/2025/Conference — NeurIPS 2025 poster_

### Official Review · Reviewer_gigz · 2025-06-15

**Clarity:** 3
**Significance:** 3
**Originality:** 3
**Rating:** 5
**Confidence:** 4

**Summary:**

This paper introduces the Next Frame Prediction Attack (NFPA), a semantic-level watermark removal attack for AI-generated images. Unlike existing pixel-level attacks that modify individual pixels, NFPA formulates watermark removal as a video generation task by treating the watermarked image as an initial frame and generating a semantically coherent "next frame" that preserves visual content while removing the watermark. The method leverages a pre-trained text-to-image diffusion model in a zero-shot manner, using DDIM inversion, optical flow transformations, and frame-attention mechanism. Experiments on eight watermarking schemes show the proposed method outperforms nine baseline attacks while maintaining image quality.

**Questions:**

- How does NFPA perform on watermarked images generated by models other than Stable Diffusion? Would the method work on naturally captured images with post-processing watermarks?
- The paper tests three basic camera motions (x-axis, y-axis, xy-axis), but how sensitive is the attack to the choice of motion pattern? Could adversarial motion patterns further improve performance?
- Given the effectiveness of NFPA, what potential defenses could watermarking schemes adopt? Could temporal consistency constraints be incorporated into watermark design?
- How can we quantitatively measure the semantic preservation beyond FID and CLIP scores? Are there cases where NFPA significantly alters image semantics?

**Ethical Concerns:**

["NO or VERY MINOR ethics concerns only"]

**Final Justification:**

I have read the rebuttal and my questions have been answered.  I increased my rating to accept.

**Limitations:**

Yes

**Quality:**

3

**Strengths And Weaknesses:**

Strengths:

- It’s an interesting idea to frame watermark removal as a next-frame prediction task. This semantic-level approach may address some limitations of existing pixel-level attacks.
- The paper evaluates against 8 different watermarking methods (both pixel-level and semantic-level) and compares with 9 attack baselines, providing thorough experimental validation.
- The proposed method combines DDIM inversion, optical flow optimization, and frame-attention mechanism.

Weaknesses:

- The paper lacks theoretical justification for why the next-frame prediction approach should effectively remove semantic watermarks. The connection between temporal consistency and watermark removal could be better explained.
- NFPA requires ~1.3 seconds per image, which is significantly slower than simple distortion attacks. The paper doesn't adequately discuss scalability for large-scale attacks.
- The method introduces several hyperparameters (optical flow bounds, inference steps, etc.) but provides limited analysis of sensitivity to these choices beyond brief ablation studies.
- Only evaluated on MS-COCO dataset with Stable Diffusion-generated images. No evaluation on real-world watermarked images or other generation models.

---

> ### Author Rebuttal · Authors · 2025-07-31
>
> We sincerely thank you for your thorough review and valuable feedback on our paper. We are especially grateful for your recognition of our **method design** and the **comprehensive experimental validation**. Below, we provide detailed responses to your concerns and questions:
>
> ## **W1: Theoretical Analysis**
> Thank you for your insightful examination of our theoretical motivation. Our proposed NFPA innovatively formulates the task of semantic watermark removal as a *next-frame prediction* problem. This formulation is grounded in the *inherent semantic continuity and subtle variations across temporal sequences in video*. Specifically, semantic watermarks often rely on particular embedding patterns in the high-dimensional feature space. NFPA leverages the fact that adjacent frames in a video exhibit strong semantic consistency with only slight pixel-level differences.
>
> By constructing an optical flow field matrix, we introduce fine-grained and controllable perturbations to the latent representations of the image. Coupled with a *frame attention mechanism*, this ensures that the generated next-frame image remains semantically coherent with the original watermarked input. These small but strategic semantic perturbations allow the output image to maintain its core semantic structure and perceptual quality while *escaping the feature subspace where the watermark was originally embedded*. In essence, while the semantic content remains intact, the high-dimensional feature distribution is subtly altered, thereby evading watermark detectors.
>
> This approach achieves a critical balance between *semantic consistency* and *perturbation effectiveness*, demonstrating that semantic changes can effectively dismantle embedded watermarks without visibly altering the image. We will elaborate more rigorously on the link between temporal consistency and watermark removal in the revised version.
>
> ---
>
> ## **W2: Scalability of Large-Scale Attacks**
> Thank you for your attention to the efficiency and scalability of our method under large-scale attack scenarios. For semantic-level watermarks, simple and fast pixel-level distortions are generally ineffective. The value of NFPA lies in its ability to remove such *robust watermarks*, which inherently requires modest computational resources.
>
> Regarding scalability, *NFPA*, as a deep learning-based model, exhibits substantial parallelization potential. It can be efficiently **batch-processed** on GPUs, significantly reducing the total runtime when handling large volumes of images. Thus, the average *1.3 seconds per image* runtime reflects a reasonable trade-off for its high attack efficacy, and the method has *ample room for optimization* through parallel processing in practical deployments.
>
> ---
>
> ## **W3: Hyperparameter Analysis**
> We appreciate your suggestion to provide a more detailed sensitivity analysis of our hyperparameters. We have now expanded our analysis as follows:
>
> 1. **Base Model Selection**: Our experiments show that NFPA maintains *robust watermark removal performance* across different base models (e.g., SD-v1.4 ~ SD-v2.1). Although larger models yield slightly better image quality, this highlights that NFPA is **insensitive to the choice of base model**, suggesting its continued effectiveness and generalizability even as diffusion models evolve.
>
> 2. **Frame Attention Mechanism**: Ablation studies reveal that removing this mechanism significantly degrades image quality, underscoring its **critical role in maintaining inter-frame semantic coherence** during attacks. It ensures that while watermarks are removed, the original semantic structure of the image is preserved — achieving a strong balance between attack strength and visual fidelity.
>
> 3. **Number of Inference Steps**: Increasing denoising steps improves image quality but may slightly reduce attack effectiveness. This is because finer denoising and more faithful reconstruction bring the predicted image closer to the original at the pixel level, potentially *weakening the semantic-level perturbation* introduced by NFPA. Thus, we set the default number of steps to 10, as a balanced trade-off between removal performance and perceptual quality.
>
> 4. **Maximum Optical Flow Search Range (δ)**: We evaluate this parameter in Appendix D. When δ = 0, no motion is introduced, and NFPA degrades into a reconstruction-based attack, resulting in poor removal performance. As δ increases, it allows more freedom to explore the feature space and decouple watermark signals, albeit with a potential drop in visual quality. This flexibility enables a controllable trade-off between detection evasion and perceptual distortion.
>
> These detailed analyses will be incorporated in the revised version to better illustrate the impact of hyperparameters on performance.
>
> ---
>
> ## **Q1&W4: Watermarks from Real Images and Other Diffusion Models**
> Thank you for raising concerns about the applicability of our method to different types of watermarked images. To address this, we conducted two additional sets of experiments:
>
> 1. **Watermarks from Other Generative Models**: We evaluate watermarked images generated by the **FLUX.1-dev** [a] model. Since no official semantic watermarking implementation exists for other diffusion architectures, we use post-processing watermarking schemes for validation. The results clearly demonstrate that **NFPA remains highly effective**, even when attacking watermarks from novel diffusion architectures. The results are as follows (TPR@1%FPR ↓):
>
> | Method | DwtDct | RivaGAN | SSL | StegaStamp | Avg. |
> |:------:|:------:|:-------:|:---:|:-----------:|:----:|
> | None   | 0.86   | 1.00    |1.00 |    1.00     | 0.97 |
> | **NFPA (ours)** | **0.01** | **0.14** | **0.15** | **0.01** | **0.08** |
>
> 2. **Real-World Watermarked Images**: We further validate NFPA on *real-world photographs* from the **MIRFLICKR** dataset. We randomly sampled 1,000 real photos and embedded watermarks using four conventional methods. NFPA effectively reduced the average watermark detection rate to **0.08**. The results are as follows (TPR@1%FPR ↓):
>
> | Method | DwtDct | RivaGAN | SSL | StegaStamp | Avg. |
> |:------:|:------:|:-------:|:---:|:-----------:|:----:|
> | None   | 0.79   | 1.00    |1.00 |    1.00     | 0.95 |
> | **NFPA (ours)** | **0.01** | **0.14** | **0.16** | **0.02** | **0.08** |
>
> These results strongly support that NFPA is both **adaptive across various generative architectures** and **practical in real-world scenarios**.
>
> ---
>
> ## **Q2: Complex Motion Models**
> Thank you for your comments on our choice of motion models and their potential influence on attack performance. We test three basic camera translation motions (X, Y, and XY axes). Regardless of the motion mode, NFPA *consistently outperforms state-of-the-art methods*, indicating **robustness to motion model choices**.
>
> We also added a new experiment using a **zoom-based motion model**, and the results are as follows (TPR@1%FPR ↓):
>
> | Method | DwtDct | RivaGAN | SSL | StegaStamp | TreeRing | StableSignature | RingID | GaussianShading | Avg. |
> |:------:|:------:|:-------:|:---:|:-----------:|:--------:|:----------------:|:------:|:----------------:|:----:|
> | **NFPA (zoom)** | 0.01 | 0.08 | 0.08 | 0.18 | 0.11 | 0.00 | 0.01 | 0.00 | 0.06 |
>
> These results further confirm NFPA’s *motion-model-insensitive design*. We recognize that more complex or dynamic motion models could unlock new attack potentials. Exploring such models is part of our future research.
>
> ---
>
> ## **Q3: Watermark Defenses**
> We appreciate your insightful question regarding possible defense strategies against NFPA. The idea of integrating temporal consistency into watermark design is particularly promising. Potential defenses could include:
>
> 1. **Enhancing Semantic Resilience**: Watermarks could be embedded in *high-level semantic features*, rather than shallow textures or frequency components. This would make any semantic-level perturbation attempt inevitably affect the image content in a detectable way.
>
> 2. **Temporal Consistency Embedding**: Designing watermarks that *persist through semantic shifts or frame predictions* could significantly challenge attacks like NFPA. For example, encoding watermarks to evolve predictably across semantic transitions would require attackers to break temporal coherence, making attacks more detectable.
>
> This could involve adversarial training to make watermark extraction models resilient to subtle semantic perturbations. We believe this is a highly valuable direction for building **next-generation robust watermarking schemes**.
>
> ---
>
> ## **Q4: Perceptual Quality Evaluation**
> Thank you for your thoughtful discussion on evaluating semantic fidelity. Beyond **FID** and **CLIP scores**, we incorporated two additional metrics:
>
> - *Q-align* [b]: A large multimodal model-based metric aligned with human perceptual judgments. It provides a better approximation of human visual assessment.
> - *Cosine Similarity (CosSim)*: Based on *SimCLR* [c] representations, this metric measures the semantic similarity between the original and attacked image, providing insight into whether the semantic content is preserved.
>
> | Method | None | JPEG | Crop | Blur | Noise | ColorJitter | Rotation | VA | DA | MSAA | NFPA |
> |-|-|-|-|-|-|-|-|-|-|-|-|
> | **Q-align ↑** | 4.02 | 3.48 | 3.20 | 2.19 | 2.07 | 2.08 | 3.07 | 3.96 | 4.10 | 1.97 | **3.88** |
> | **CosSim ↑** | 0.30 | 0.27 | 0.29 | 0.30 | 0.29 | 0.29 | 0.30 | 0.30 | 0.29 | 0.30 | **0.30** |
>
> These results demonstrate that **NFPA achieves high semantic and perceptual fidelity**, validating our core motivation: using *next-frame prediction* to effectively remove watermarks *without compromising image quality or semantics*.
>
> ---
>
> **References**
>
> [a] https://huggingface.co/black-forest-labs/FLUX.1-dev
>
> [b] Q-Align: Human-Aligned Quality Estimation for Text-to-Image Generation, CVPR 2024
>
> [c] A Simple Framework for Contrastive Learning of Visual Representations, ICML 2020

---

> > ### Comment · Reviewer_gigz · 2025-08-01
> > **Thank you for the rebuttal**
> >
> > I have read the rebuttal and my questions have been answered.

---

> > > ### Author Response · Authors · 2025-08-02
> > >
> > > Dear Reviewer gigz,
> > >
> > > Thank you very much for your careful review and for taking the time to read our rebuttal. We have made every effort to address your concerns and have revised the manuscript accordingly. We hope that our responses have satisfactorily clarified all your questions. With these updates, we kindly ask you to consider the possibility of revisiting your rating.
> > >
> > > Please feel free to reach out if you have any further questions. Once again, we sincerely appreciate your time and valuable feedback.
> > >
> > > Sincerely,
> > >
> > > Authors

---

### Official Review · Reviewer_g6pS · 2025-07-01

**Clarity:** 2
**Significance:** 2
**Originality:** 3
**Rating:** 4
**Confidence:** 3

**Summary:**

This paper studied the watermark removal attack problem. Specifically, the authors pointed out that current methods might fall short in terms of semantic-level watermark removal, and to address this issue, they proposed the Next Frame Prediction Attack (NFPA) aimed at semantic-level removal. The NFPA models watermark removal as a video generation task where the goal is to generate the next frame that does not carry the watermark. Experiments were conducted to demonstrate the effectiveness of their proposed method.

**Questions:**

Some major questions/issues are listed below:

- My main question concerns whether a semantic-watermark removal attack is truly necessary, which relates to the foundation of this paper. Typically, watermarks involve embedding certain signals (e.g., imperceptible noise) into images that can later be extracted to verify ownership when we encounter similar images. Traditional attacks against watermarks essentially aim to remove those embedded signals through techniques like cropping, rotating, or adding noise.

However, there's an important constraint: attacks typically cannot be too aggressive, as substantially altering the images would render the original content meaningless. This makes me curious about the practical need for semantic-watermark removal attacks, since they fundamentally change the semantic meaning of images, making them significantly different from the originals. This approach seems to diverge from typical real-world watermark applications. After all, if we change an image's semantic meaning entirely, doesn't it become questionable to make copyright claims on the resulting image?

- In the experimental evaluation section, I noticed metrics focus on FID and CLIP scores rather than PSNR and SSIM. It would be helpful to explain why these traditional image quality metrics were excluded, as they're standard in measuring preservation of image fidelity.

- I would like to see results using a wider variety of diffusion models to demonstrate the robustness of your approach across different frameworks.

Overall, I find the concept interesting but would appreciate more discussion on these aspects to fully understand the practical applications and limitations of semantic watermark removal.

**Ethical Concerns:**

["NO or VERY MINOR ethics concerns only"]

**Final Justification:**

I think this paper is around the borderline, might be slightly leaning towards acceptance. I am not buying the semantic concept overall because I feel that the semantic watermark is not practical at all. For instance, an artist may not want to sacrifice or change the semantics of his/her work to add a watermark. After all, there are invisible watermarks. In this sense, I do feel this paper's practicality is very limited.

**Limitations:**

Check above

**Quality:**

2

**Strengths And Weaknesses:**

Significance: Watermark-related research problems are of potentially great significance in the generative AI era.

Clarity: I think the paper does not clearly discuss and distinguish between classical pixel-watermarks and semantic watermarks.

Quality: I think the overall writing and the experiments presented are OK.

Originality: The idea of casting the watermark removal process as video generation is novel, at least to me.

---

> ### Author Rebuttal · Authors · 2025-07-31
>
> We sincerely thank you for the detailed and thoughtful review of our paper. We truly appreciate your recognition of the **novelty**, **significance**, and **overall quality** of our work. Below, we provide detailed responses to your insightful comments and concerns:
>
> ---
>
> ## **Q1: The Necessity of Semantic-level Attacks**
>
> Thank you for raising this fundamental question. We fully understand your concern that *attacks should avoid damaging the semantic integrity of the image*. In fact, one of the core objectives of our proposed method is to achieve **watermark removal with minimal semantic shift**. As shown in *Figure 4* of the main paper, the outputs generated by our method are *visually near-equivalent* to the original watermarked images, indicating that **the image content remains essentially intact**.
>
> More importantly, **semantic-level removal attacks are necessary**. Recent advances have introduced a new class of *semantic-level watermarking schemes* (e.g., *TreeRing*, *RingID*) that embed signals into the *semantic latent space* of AIGC images by altering the noise or structural priors. These schemes are designed to be significantly more robust than classical pixel-based watermarks. However, without corresponding semantic-level attacks, there is currently no principled way to evaluate their real-world security. This could lead to *false confidence* in these watermarking methods, posing risks for applications like deepfake detection and content provenance.
>
> Our work aims to fill this gap by proposing the **first semantic-level watermark removal attack**. We formulate the problem as a next-frame prediction task, enabling *fine-grained semantic manipulation* while maintaining perceptual consistency through frame attention. Moreover, we would like to emphasize that our goal is to **expose vulnerabilities in existing image watermarking systems**, which can **inspire the development of stronger and more resilient watermarking techniques**. NFPA also serves as a valuable *security evaluation tool*, guiding the design of future watermarking methods and contributing to a more trustworthy AI ecosystem.
>
> ---
>
> ## **Q2: Image Quality Metrics**
>
> Thank you for pointing out the importance of evaluation metrics. We fully agree that *PSNR and SSIM are standard and meaningful for pixel-level fidelity assessments*. However, as you rightly observed, our focus is on *semantic-level watermark removal*, which necessitates a shift in the evaluation perspective:
>
> - *PSNR and SSIM* are designed to measure *pixel-wise reconstruction quality*, suitable for tasks like image compression or denoising where spatial alignment is preserved;
> - In contrast, our method (NFPA) performs **semantic transformations** that may introduce spatial variations (e.g., pixel rearrangements), even though the **semantic content and perceptual quality remain intact**;
> - In such scenarios, *pixel-aligned metrics can produce misleadingly low scores*, failing to reflect true semantic consistency.
>
> To address this concern and further support our claims, we have introduced two additional evaluation metrics:
>
> - *Q-align* [a]: A large multimodal model-based metric aligned with human perceptual judgments. It provides a better approximation of human visual assessment.
> - *Cosine Similarity (CosSim)*: Based on *SimCLR* [b] representations, this metric measures the semantic similarity between the original and attacked image, providing insight into whether the semantic content is preserved.
>
> | Method | None (w/o attack)| JPEG | Crop | Blur | Noise | ColorJitter | Rotation | VA | DA | MSAA | NFPA (ours) |
> |:------:|:----:|:----:|:----:|:----:|:-----:|:-----------:|:--------:|:--:|:--:|:----:|:------------:|
> | Q-align↑ | 4.02 | 3.48 | 3.20 | 2.19 | 2.07 | 2.08 | 3.07 | 3.96 | 4.10 | 1.97 | 3.88 |
> | CosSim↑ | 0.30 | 0.27 | 0.29 | 0.30 | 0.29 | 0.29 | 0.30 | 0.30 | 0.29 | 0.30 | 0.30 |
>
> These results demonstrate that **NFPA achieves high perceptual and semantic fidelity**, without compromising watermark removal effectiveness.
>
> ---
>
> ## **Q3: Evaluation on more diffusion architectures**
>
> We appreciate your suggestion to extend the experimental scope, as it is crucial to validate the **robustness and generality** of our method. In response, we have conducted additional experiments to evaluate NFPA’s performance on images generated by the *FLUX.1-dev* [c] diffusion model. Since some watermarking schemes lack implementations for alternative diffusion architectures, we focus on *post-hoc watermarking methods* to test cross-model transferability of our attack.
>
> The results are as follows (TPR@1%FPR ↓):
>
> | Method         | DwtDct | RivaGAN | SSL  | StegaStamp | Avg. |
> |----------------|--------|---------|------|-------------|------|
> | None (w/o attack) | 0.86   | 1.00    | 1.00 | 1.00        | 0.97 |
> | **NFPA (ours)**       | **0.01**   | **0.14**    | **0.15** | **0.01**        | **0.08** |
>
> These results show that **NFPA remains highly effective even on images generated by a novel diffusion model**, demonstrating strong transferability and adaptability across different generative backbones.
>
> ---
>
> ## **W1: Clarifying the Distinction Between Pixel-Level and Semantic-Level Watermarking**
>
> We sincerely appreciate your insightful comment. We fully agree that a clearer and more explicit differentiation between *pixel-level* and *semantic-level* watermarking will enhance the reader's understanding of the distinct threat models, attack surfaces, and potential vulnerabilities associated with each. In the final version of our paper, we will include the following refined description:
>
> - **Pixel-level watermarking methods** (e.g., *DwtDct*, *RivaGAN*, *SSL*, *StableSignature*) embed imperceptible ownership signals directly into the image’s **pixel or frequency domain**, often via spatial perturbations, quantization, or frequency modulation. These techniques are typically vulnerable to classical distortions such as compression, cropping, or noise injection.
>
> - **Semantic-level watermarking methods** (e.g., *TreeRing*, *RingID*, *GaussianShading*, *StegaStamp*) typically operate in the latent space of diffusion models. They encode ownership by perturbing **semantic features**, making them more resilient to low-level attacks but potentially more susceptible to *semantic-space manipulations*, like our proposed method.
>
> This distinction also highlights the necessity of our attack, where semantic-level watermarking introduces new robustness challenges and requires more advanced removal strategies to verify their robustness.
>
> ---
>
> Again, thank you for your insightful feedback and deep engagement with our work. Your suggestions have significantly helped us **clarify the motivation, strengthen the empirical foundation, and refine the overall presentation** of our paper. We have incorporated the necessary extensions and explanations in the revised version, and we hope the improved clarity and rigor will better communicate our contributions.
>
> **References**
>
> [a] Wu et al. *Q-ALIGN: Teaching LMMs for Visual Scoring via Discrete Text-defined Levels.* ICML 2024.
> [b] Chen et al. *A Simple Framework for Contrastive Learning of Visual Representations.* ICML 2020.
> [c] https://huggingface.co/black-forest-labs/FLUX.1-dev

---

> > ### Author Response · Authors · 2025-08-06
> >
> > Dear Reviewer g6pS,
> >
> > Thank you very much for your thoughtful suggestions. We have clarified the necessity of semantic-level attacks, incorporated additional image quality metrics, and demonstrated the generalization ability of our method on a new diffusion model (FLUX.1-dev). We hope these revisions adequately address your concerns.
> >
> > If our clarifications are satisfactory, we would be sincerely grateful if you would consider revising your score.
> >
> > Should you have any further questions or concerns, please do not hesitate to reach out to us. Once again, we deeply appreciate the time and effort you have dedicated to reviewing our work.
> >
> > Sincerely,
> >
> > The Authors

---

> > > ### Comment · Reviewer_g6pS · 2025-08-07
> > >
> > > I appreciate the authors' detailed feedback. I still do not fully agree on the arguments you provided regarding the necessity of semantic watermarks. However, I am willing to increase my score to weak acceptance considering the overall quality of this paper.

---

> > > > ### Author Response · Authors · 2025-08-07
> > > >
> > > > Dear Reviewer g6pS,
> > > >
> > > > Thank you sincerely for your thoughtful feedback and for your decision to raise the score. We truly appreciate your acknowledgment of the overall quality of our work, which is highly encouraging to us.
> > > >
> > > > Best regards,
> > > >
> > > > The Authors

---

### Official Review · Reviewer_seke · 2025-07-05

**Clarity:** 3
**Significance:** 3
**Originality:** 2
**Rating:** 4
**Confidence:** 4

**Summary:**

This paper proposes Next Frame Prediction Attack (NFPA), the first semantic-level attack specifically designed to remove watermarks embedded in AI-generated images. NFPA formulates watermark removal as a next-frame prediction problem inspired by video generation: it treats the watermarked image as an initial frame and synthesizes an unwatermarked next frame by subtly modifying the image’s semantic structure. The method leverages a zero-shot, tuning-free framework based on a pre-trained text-to-image diffusion model, performing DDIM inversion to obtain the latent representation of the input and applying an optimized optical flow transformation in the latent space to disrupt watermark signals while preserving perceptual fidelity. The authors conduct extensive experiments on eight state-of-the-art watermarking schemes, covering both pixel-level and semantic-level methods, and compare NFPA with nine established attack baselines, including distortion, regeneration, and adversarial attacks. Results demonstrate that NFPA consistently reduces watermark detectability while maintaining high image quality measured by FID and CLIP scores.

**Questions:**

Please see the weakness section for detailed concerns that require clarification.

**Ethical Concerns:**

["NO or VERY MINOR ethics concerns only"]

**Final Justification:**

I appreciate the authors’ detailed and constructive rebuttal, which addressed several of my initial concerns. They added missing baselines such as CtrlRegen+, explored simpler alternatives like translation and inpainting, and extended comparisons to later SVD frames, showing meaningful improvements.

However, I still believe that SVD’s effectiveness could be further improved by tuning the "noise-aug-strength" parameter, which controls the degree of semantic drift in generation and can act as an implicit watermark removal strength knob. Additionally, the semantic shifts introduced by NFPA (as seen in Supplementary Fig. 1) raise questions about its alignment with the “next-frame” prediction framing and warrant further discussion.

Nonetheless, considering that several concerns were meaningfully addressed and the method is likely to be of interest to the community, I have raised my score to borderline accept.

**Limitations:**

Yes

**Paper Formatting Concerns:**

There are no concerns regarding formatting.

**Quality:**

3

**Strengths And Weaknesses:**

$\textbf{Strengths}$

1 -- The paper proposes a novel method for removing semantic-level watermarks by treating the problem as a next-frame prediction task. This approach goes beyond traditional pixel-level attacks and introduces a new perspective for attacking advanced watermarking techniques.

2 -- The paper presents a well-designed methodology by adapting a pre-trained text-to-image diffusion model for next-frame prediction. The authors combine DDIM inversion with an optimized optical flow transformation in the latent space to generate an unwatermarked image that remains visually and semantically similar to the original.

3 -- The proposed attack works in a black-box setting without requiring any information about the watermark or the detector, and it does not rely on additional data or queries. This makes the attack practical and realistic for situations where the attacker has limited knowledge or resources.

$\textbf{Weaknesses}$

1 -- The paper does not compare NFPA with recent work [1] showing that controllable regeneration from clean noise can remove even semantic-level watermarks, despite claiming that existing semantic watermarking techniques are robust against such attacks; this omission weakens the comparative analysis.

2 -- A relatively simpler approach would be to use off-the-shelf image-to-video generation modules, such as Stable Diffusion-based video models, which can directly generate temporally coherent next frames without requiring the complex inversion and optical flow steps proposed by the authors. However, the paper does not explore or compare these simpler alternatives, leaving it unclear whether NFPA’s complexity is necessary.

3 -- The next-frame images produced by NFPA appear primarily as translated versions of the originals, suggesting that simple image translation alone might remove much of the watermark. The paper should compare NFPA with basic translation attacks, potentially combined with diffusion-based inpainting to reconstruct missing regions created by the translation.

4 --  The attack design is limited to translation-based motion and does not consider other common camera transformations, such as zoom or rotation around the subject, which could lead to more effective or diverse watermark removal strategies.

5 -- While quantitative metrics and qualitative examples are provided, the paper lacks a user study or detailed perceptual evaluation to confirm whether the subtle semantic changes introduced by NFPA degrade the recognizability or utility of attacked images.

[1] - IMAGE WATERMARKS ARE REMOVABLE USING CONTROLLABLE REGENERATION FROM CLEAN NOISE (ICLR, 2025)

---

> ### Author Rebuttal · Authors · 2025-07-31
>
> We sincerely thank you for your detailed and insightful review. We are glad that you recognized the **novelty** of our method and its **practical value in black-box, data-free, and query-free** scenarios. Below, we provide detailed responses to each of your valuable concerns:
>
> ---
>
> ## **W1: Baseline Comparison: CtrlRegen+**
>
> We greatly appreciate your suggestion to include *CtrlRegen+* [1] as a key baseline. Following your recommendation, we have conducted a thorough evaluation and **commit to including the complete results and quantitative analysis** in the final version.
>
> We faithfully follow the default hyperparameter (`step=0.5`) in the official open-source implementation. The results are as follows (TPR@1%FPR ↓):
>
> | Method         | DwtDct | RivaGAN | SSL  | StegaStamp | TreeRing | StableSignature | RingID | GaussianShading | Avg. |
> |----------------|--------|---------|------|-------------|----------|------------------|--------|------------------|------|
> | CtrlRegen+     | 0.01   | 0.02    | 0.03 | 0.36        | 0.73     | 0.00             | 0.96   | 1.00             | 0.39 |
> | **NFPA (ours)** | **0.01**   | **0.13**    | **0.09** | **0.02**        | **0.07**     | **0.00**             | **0.02**   | **0.00**             | **0.04** |
>
> Clearly, *CtrlRegen+* shows **limited effectiveness** on several watermarking methods, particularly *RingID (0.96)* and *GaussianShading (1.00)*. In contrast, **our NFPA achieves strong generalizability across all watermark types**.
>
> Furthermore, CtrlRegen+ requires training two distinct modules (semantic and spatial controllers), incurring **significantly higher training costs**. In contrast, **NFPA is a zero-shot method**, leveraging pretrained diffusion models for direct next-frame prediction without any fine-tuning or additional training. This makes **NFPA more deployable and practical in real-world attack settings**.
>
> ---
>
> ## **W2: Baseline Comparison: Video Generation Model**
>
> Thank you for the insightful suggestion. To assess this idea, we evaluated *Stable Video Diffusion* (SVD) [a]. We used its second generated frame as the next-frame image. The results are as follows (TPR@1%FPR ↓):
>
> | Method         | DwtDct | RivaGAN | SSL  | StegaStamp | TreeRing | StableSignature | RingID | GaussianShading | Avg. |
> |----------------|--------|---------|------|-------------|----------|------------------|--------|------------------|------|
> | SVD            | 0.10   | 0.53    | 0.51 | 1.00        | 0.91     | 0.10             | 0.99   | 0.76             | 0.61 |
> | **NFPA (ours)** | **0.01**   | **0.13**    | **0.09** | **0.02**        | **0.07**     | **0.00**             | **0.02**   | **0.00**             | **0.04** |
>
> SVD performs substantially worse than NFPA, likely due to two key limitations:
>
> 1. It lacks explicit mechanisms for watermark removal;
> 2. Its inference time is high (∼31 seconds per image), making it impractical for real-time attacks.
>
> By contrast, **NFPA leverages flow-guided latent perturbation to remove watermark**, achieving a better trade-off between **efficiency and effectiveness**.
>
> ---
>
> ## **W3: Baseline Comparison: Translation + Inpainting**
>
> As suggested, we implemented two baselines for comparison:
>
> - *Translation only*: We follow the NFPA settings to randomly translate the image along the x/y axis using the `ImageChops.offset(.)` function, with a translation range of [-40, 40] pixels;
> - *Translation + Inpainting*: shifted images are completed using the *stable-diffusion-2-inpainting* [b] model.
>
> The results are as follows (TPR@1%FPR ↓):
>
> | Method                | DwtDct | RivaGAN | SSL  | StegaStamp | TreeRing | StableSignature | RingID | GaussianShading | Avg. |
> |------------------------|--------|---------|------|-------------|----------|------------------|--------|------------------|------|
> | Translation only            | 0.03   | 1.00    | 1.00 | 0.17        | 0.38     | 1.00             | 0.31   | 0.01             | 0.49 |
> | Translate + Inpainting | 0.00   | 0.36    | 0.56 | 0.00        | 0.13     | 0.01             | 0.01   | 0.00             | 0.13 |
> | **NFPA (ours)**        | **0.01**   | **0.13**    | **0.09** | **0.02**        | **0.07**     | **0.00**             | **0.02**   | **0.00**             | **0.04** |
>
> We summarize two key points of analysis:
> 1.  **Naive translation fails to remove most watermarks**, with an average detection rate of 0.49;
> 2.  *Translation + Inpainting* improves slightly but **still underperforms NFPA and lacks a clear watermark removal strategy.**
>
> These results further justify our formulation of the task as **next-frame prediction** rather than simple spatial augmentation.
>
> ---
>
> ## **W4: More Camera Transformations**
>
> Thank you for highlighting this point. We test three basic camera translation motions (X, Y, and XY axes). Regardless of the motion mode, NFPA consistently outperforms state-of-the-art methods, indicating **robustness to motion model choices**. We have extended NFPA to include **zoom-based camera motion** and compared it with the original translation variant. The results are as follows (TPR@1%FPR ↓):
>
> | Method          | DwtDct | RivaGAN | SSL  | StegaStamp | TreeRing | StableSignature | RingID | GaussianShading | Avg. |
> |------------------|--------|---------|------|-------------|----------|------------------|--------|------------------|------|
> | NFPA (zoom)      | 0.01   | 0.08    | 0.08 | 0.18        | 0.11     | 0.00             | 0.01   | 0.00             | 0.06 |
> | NFPA (xy-axis)   | 0.01   | 0.13    | 0.09 | 0.02        | 0.07     | 0.00             | 0.02   | 0.00             | 0.04 |
>
> The results further verify that **NFPA remains robust across different types of camera motion**, supporting its flexibility and generalization. We plan to explore more advanced transformations as part of our future work.
>
> ---
>
> ## **W5: Perceptual Quality Evaluation**
>
> We appreciate your focus on visual quality. To further evaluate perceptual and semantic preservation, we introduce two complementary metrics:
>
> - *Q-align* [c]: A large multimodal model-based metric aligned with human perceptual judgments. It provides a better approximation of human visual assessment.
> - *Cosine Similarity (CosSim)*: Based on *SimCLR* [d] representations, this metric measures the semantic similarity between the original and attacked image, providing insight into whether the semantic content is preserved.
>
> | Method | None (w/o attack) | JPEG | Crop | Blur | Noise | ColorJitter | Rotation | VA | DA | MSAA | NFPA (ours) |
> |:------:|:----:|:----:|:----:|:----:|:-----:|:-----------:|:--------:|:--:|:--:|:----:|:------------:|
> | **Q-align↑** | 4.02 | 3.48 | 3.20 | 2.19 | 2.07 | 2.08 | 3.07 | 3.96 | 4.10 | 1.97 | **3.88** |
> | **CosSim↑** | 0.30 | 0.27 | 0.29 | 0.30 | 0.29 | 0.29 | 0.30 | 0.30 | 0.29 | 0.30 | **0.30** |
>
> These results show that **NFPA achieves high perceptual quality and semantic fidelity** without compromising on watermark removal. In the final version, we will include more perceptual examples and plan to incorporate user studies as a future extension.
>
> ---
>
> We thank you again for your constructive feedback, which helped us substantially improve our work. We have added key baselines, performed in-depth ablation studies, and provided more comprehensive justifications based on your input. We believe these enhancements significantly strengthen the completeness and persuasiveness of our paper.
>
> **References**
>
> [1] *Image Watermarks Are Removable Using Controllable Regeneration from Clean Noise*, ICLR 2025
>
> [a] https://huggingface.co/stabilityai/stable-video-diffusion-img2vid
>
> [b] https://huggingface.co/stabilityai/stable-diffusion-2-inpainting
>
> [c] *Q-Align: Human-Aligned Quality Estimation for Text-to-Image Generation*, CVPR 2024
>
> [d] *A Simple Framework for Contrastive Learning of Visual Representations*, ICML 2020

---

> > ### Comment · Reviewer_seke · 2025-08-05
> >
> > Thank you for the additional experiments.
> >
> > Could the authors clarify why only the second frame from Stable Video Diffusion (SVD) was used in the comparison? Since early frames typically introduce limited semantic change, this choice may underestimate the potential of video-based approaches for watermark removal. Evaluating later frames, which tend to exhibit greater semantic drift, would provide a more balanced and representative comparison. This is particularly relevant given prior work (e.g., Robust Watermarking Using Generative Priors Against Image Editing: From Benchmarking to Advances, ICLR25) has shown that image-to-video generation can significantly degrade a wide range of watermarking methods.

---

> > > ### Author Response · Authors · 2025-08-06
> > >
> > > Thank you for your valuable comments. We greatly appreciate your feedback. Regarding the choice of using the *second frame* from SVD for comparison, we would like to provide the following clarifications and justifications:
> > >
> > > **1. Content Integrity Constraint:**
> > > The core challenge of image watermark removal attacks lies in effectively removing the watermark while ensuring *high integrity of the image content*. The attacked image must remain visually and semantically consistent with the original watermarked image. In video generation tasks, adjacent frames exhibit strong spatiotemporal coherence, thus the second frame generated by SVD is visually nearly equivalent to the original watermarked image, which serves as the first frame, satisfying the content integrity constraint. However, later frames inevitably introduce significant semantic drift, which violates the fundamental premise of removal attacks.
> > >
> > > **2. Fair Comparison with NFPA:**
> > > Our NFPA method models watermark removal as a "next-frame prediction" task. To ensure fairness, when using SVD as a baseline, we follow the same setting by taking the watermarked image as input and extracting the generated second frame as the next-frame prediction. Experimental results clearly show that under the "next-frame prediction" paradigm, SVD, which lacks an explicit watermark removal mechanism, performs significantly worse than NFPA. This strongly validates the uniqueness and effectiveness of our method on this specific task.
> > >
> > > Furthermore, we agree with your point that evaluating later frames of SVD may show stronger watermark removal performance. However, as emphasized, this improvement comes at the cost of sacrificing content integrity, which contradicts the objective of watermark removal attacks. Our NFPA method achieves a better balance between watermark removal effectiveness and image quality through a carefully designed optical flow transformation strategy and frame attention mechanism. The visual comparisons in Figure 4 further support that our approach can effectively remove watermarks while ensuring the attacked image is visually almost indistinguishable from the original.

---

> ### Comment · Reviewer_seke · 2025-08-06
>
> Thank you for the clarification. However, I respectfully disagree with the clarification points.
>
> While the authors argue that later frames introduce semantic drift, SVD is explicitly trained to preserve semantic consistency across time, and even frames up to 20-30 typically maintain the object-level semantics. Notably, the proposed NFPA method itself introduces substantial semantic changes, as visible in Supplementary Figure 1, that likely similar to what later SVD frames produce.
>
> Further regarding fair comparison, I understand that next-frame prediction is the core motivation of NFPA. However, the visual examples in the supplementary material (e.g., Figure 1) clearly show that the semantic and spatial shifts produced by NFPA go well beyond what would be expected from a strict "next-frame" prediction. These changes are unlikely to arise from a single-step temporal shift and resemble the kind of gradual transformation observed across multiple frames.

---

> > ### Author Response · Authors · 2025-08-07
> >
> > Thank you for your continued thoughtful feedback. Your insights are highly valuable in helping us better position the strengths and contributions of our work.
> >
> > To address your concerns, we have conducted a thorough evaluation of SVD as a baseline and extended our analysis to include its watermark removal performance on both an intermediate frame (frame 7) and the later frame (frame 14). The results are presented in the table below (TPR@1%FPR ↓):
> >
> > | Method         | DwtDct | RivaGAN | SSL  | StegaStamp | TreeRing | StableSignature | RingID | GaussianShading | Avg. |
> > |:--------------:|:------:|:-------:|:----:|:-----------:|:--------:|:----------------:|:------:|:----------------:|:----:|
> > | SVD (2-frame)  | 0.10   | 0.53    | 0.51 | 1.00        | 0.91     | 0.10             | 0.99   | 0.76             | 0.61 |
> > | SVD (7-frame)  | 0.07   | 0.39    | 0.45 | 0.44        | 0.28     | 0.04             | 0.23   | 0.14             | 0.26 |
> > | SVD (14-frame) | 0.05   | 0.26    | 0.22 | 0.25        | 0.12     | 0.01             | 0.16   | 0.15             | 0.15 |
> > | **NFPA (ours)**    | **0.01**   | **0.13**    | **0.09** | **0.02**        | **0.07**     | **0.00**             | **0.02**   | **0.00**            | **0.04** |
> >
> > As anticipated, SVD's performance improves when using later frames. However, its overall effectiveness remains limited due to the absence of an explicit mechanism for watermark removal. In contrast, NFPA consistently achieves **state-of-the-art results** across all watermark types.
> >
> > Additionally, we would like to reiterate NFPA's significant advantage in efficiency. SVD requires approximately 31 seconds to generate a single video, which can be impractical for real-time or large-scale deployment. NFPA completes the attack in approximately 1 second, offering a compelling balance between effectiveness and efficiency.
> >
> > In summary, these expanded comparisons further demonstrate the strengths of our proposed method. NFPA, through its tailored next-frame prediction framework, delivers an effective and efficient solution for image watermark removal—achieving strong performance, maintaining content integrity, and minimizing computational overhead. We believe this highlights both the practicality and novelty of our approach.
> >
> > We sincerely appreciate your feedback once again. Your comments have been instrumental in improving our manuscript and in helping us better articulate the contributions of our work.

---

> > > ### Comment · Reviewer_seke · 2025-08-07
> > >
> > > I appreciate the authors' thorough response, and the effort put into addressing my concerns. The inclusion of later SVD frames (e.g., frame 7 and 14) provides a much more balanced comparison.
> > >
> > > Regarding the absence of an explicit watermark removal mechanism in SVD, the distance to the watermarked image can be controlled via the "noise-aug-strength" parameter in the SVD pipeline. I still believe that with careful tuning, SVD (or similar video models) could yield even stronger results. Nevertheless, as the authors point out, NFPA has advantage in terms of runtime efficiency, which is an important practical consideration.
> > >
> > > Considering that my core concerns have been partially addressed, and also taking into account feedback from other reviewers, I am raising my score. For the camera-ready version, I encourage the authors to include the new results and discuss the trade-offs between effectiveness and efficiency, as this will help clarify NFPA’s positioning for future readers.

---

> > > > ### Author Response · Authors · 2025-08-07
> > > >
> > > > Dear Reviewer seke,
> > > >
> > > > Thank you very much for your thoughtful feedback and for your willingness to raise your score. We sincerely appreciate your recognition of our efforts to address your concerns.
> > > >
> > > > As you kindly suggested, we will include the additional results in the camera-ready version and expand the discussion on the balance between effectiveness and efficiency. We believe this will help better convey the practical positioning and contributions of NFPA to the community.
> > > >
> > > > Once again, thank you for your valuable input and for helping us improve the clarity and completeness of our work.
> > > >
> > > > Sincerely,
> > > >
> > > > The Authors

---

### Official Review · Reviewer_m5Bd · 2025-07-06

**Clarity:** 3
**Significance:** 3
**Originality:** 3
**Rating:** 4
**Confidence:** 3

**Summary:**

The paper studies removing watermarks from the generated images and proposes a semantic-level removal attack named NFPA. NFPA treated the watermark removal task as a next-frame prediction task and modified the semantic structure of the watermarked image. Experiments validated its removal attack effectiveness and showed it can preserve high image quality after removal.

**Questions:**

See weakness.

**Ethical Concerns:**

["NO or VERY MINOR ethics concerns only"]

**Final Justification:**

I thank the authors for the responses, which partially alleviated my concerns. The proposed next frame prediction attack appears interesting to me, and the results show promising (particularly after addressing some baseline issues). After checking comments and responses to other reviewers, I am inclined to borderline accept this work (though the comparison with IRA does not seem quite significant, and the commonly used LPIPS was not evaluated as a metric). The additional results are supposed to be included in the revised manuscript.

**Limitations:**

See weakness.

**Paper Formatting Concerns:**

no.

**Quality:**

2

**Strengths And Weaknesses:**

Strength:

1) This paper presents a relatively novel view to remove image watermarks, ie. conducting a next frame prediction attack, by introducing temporal consistency, the proposed method is able to ensure semantic consistency. This poses a relatively new perspective to image watermark removal area.

2) This work proposed flow-based transformation strategies to simulate camera-induced motions by noise swarping, which were evaluated with different motion conditions. The module was experimentally showed effective in experiments.

3) With the TPR@1%FPR metric, the method showed competitive watermark removal performances with some existing attacks (though more evaluation metrics are suggested to be reported, see weakness).

Weakness:

1) Though some baselines, such as distortion attacks, DA and VAE-Attack were compared, to be more compelling, I suggest the authors consider more recent baselines, e.g. [1], and I will raise my score if the proposed method could still achieve competitive performance( e.g. sota).

2) Besides, for image quality measurement, it is suggested to use LPIPS [2], which can better reflect the visual fidelity wrt human visual systems.

3) In addition, it is suggested to discuss the applicability of the proposed method, whether it is able to remove some other types of watermarks (rather than T2I generative models), such as conventional image watermarks.

[1] Müller, Andreas, Denis Lukovnikov, Jonas Thietke, Asja Fischer, and Erwin Quiring. "Black-box forgery attacks on semantic watermarks for diffusion models." arXiv 2025, https://arxiv.org/abs/2412.03283

[2] Richard Zhang, et al. The Unreasonable Effectiveness of Deep Features as a Perceptual Metric. CVPR 2018

---

> ### Author Rebuttal · Authors · 2025-07-31
>
> We sincerely thank the reviewer for the thorough evaluation and insightful feedback. Your recognition of the **novelty** and **performance** of our method is truly encouraging. Below, we provide detailed responses to the three main concerns you raised:
>
> ---
>
> ## **W1: Attack Baselines: IRA**
>
> We greatly appreciate your suggestion to include the *Imprint-Removal Attack (IRA)* [1] as an additional strong baseline. Following your recommendation, we have conducted the corresponding experiments and **commit to including the complete results and quantitative analysis** in the final version.
>
> In our replication, we used SD2.1 as both the target and surrogate model, which corresponds to the white-box setting of the original IRA—effectively representing its upper-bound performance. Our results show that **NFPA outperforms IRA in average attack success rate**. Furthermore, our method exhibits clear advantages over IRA in two key aspects:
>
> ### **1. Superior Attack Stability**
>
> IRA is a typical adversarial optimization-based method that is **highly sensitive to the choice of surrogate model**. As noted in the original paper, when using SDXL as the surrogate model, IRA still fails to reduce TreeRing watermark detection below *0.64* after 50 optimization steps. In contrast, **NFPA maintains strong and consistent performance across a wide range of watermarking schemes**, making it more suitable for open-world attack scenarios.
>
> ### **2. High Efficiency and Practicality**
>
> IRA requires per-image adversarial optimization, taking **approximately 15 minutes per image**, which is computationally expensive and impractical for large-scale attacks. NFPA, on the other hand, completes watermark removal in **around 1 second per image**, making it highly deployable and scalable in practice.
>
> The comparative results are shown below (TPR@1%FPR ↓):
>
> | Method | DwtDct | RivaGAN | SSL | StegaStamp | TreeRing | StableSignature | RingID | GaussianShading | Avg. |
> |:------:|:------:|:-------:|:---:|:-----------:|:--------:|:----------------:|:------:|:----------------:|:----:|
> | IRA    | 0.02   | 0.02    |0.00 |    0.15     |  0.04    |       0.00       |  0.14  |       0.00       | 0.05 |
> | **NFPA (ours)** | **0.01** | **0.13** | **0.09** | **0.02** | **0.07** | **0.00** | **0.02** | **0.00** | **0.04** |
>
> These results clearly demonstrate that **NFPA achieves superior stability, efficiency, and practicality**, thereby strengthening the practical and academic value of our contribution.
>
> ---
>
> ## **W2: Image Quality Metrics**
>
> Thank you for suggesting the use of *LPIPS* [2] to improve perceptual evaluation. While we agree that LPIPS is a widely-used perceptual quality metric, we also recognize that it is fundamentally a *pixel-space metric*, computing L2 distances between features extracted at each pixel location. This makes it *highly sensitive to slight spatial shifts*, which may not accurately reflect performance in *semantic-level removal attack* like ours.
>
> To better capture perceptual and semantic fidelity, we consider two new metrics:
>
> - *Q-align* [a]: A large multimodal model-based metric aligned with human perceptual judgments. It provides a better approximation of human visual assessment.
> - *Cosine Similarity (CosSim)*: Based on *SimCLR* [b] representations, this metric measures the semantic similarity between the original and attacked image, providing insight into whether the semantic content is preserved.
>
> Results are summarized below:
>
> | Method | None (w/o attack) | JPEG | Crop | Blur | Noise | ColorJitter | Rotation | VA | DA | MSAA | NFPA (ours) |
> |:------:|:----:|:----:|:----:|:----:|:-----:|:-----------:|:--------:|:--:|:--:|:----:|:------------:|
> | **Q-align↑** | 4.02 | 3.48 | 3.20 | 2.19 | 2.07 | 2.08 | 3.07 | 3.96 | 4.10 | 1.97 | **3.88** |
> | **CosSim↑** | 0.30 | 0.27 | 0.29 | 0.30 | 0.29 | 0.29 | 0.30 | 0.30 | 0.29 | 0.30 | **0.30** |
>
> These results indicate that **NFPA has minimal impact on visual quality and semantic consistency**. This strongly supports our motivation of **preserving semantic integrity through next-frame prediction**.
>
> ---
>
> ## **W3: On the Applicability to Conventional Image Watermarks**
>
> We appreciate your attention to the applicability of NFPA beyond T2I watermarks. As discussed in Section 4.2 of our paper, we systematically evaluated NFPA against *four representative conventional watermarking methods* that are *not derived from generative models*: DwtDct, RivaGAN, SSL, and StegaStamp.
>
> To further validate its practicality, we applied these watermarks to *1,000 real-world photos* randomly sampled from the MIRFLICKR dataset and then evaluated NFPA’s effectiveness on this benchmark. The results (TPR@1%FPR ↓) are as follows:
>
> | Method | DwtDct | RivaGAN | SSL | StegaStamp | Avg. |
> |:------:|:------:|:-------:|:---:|:-----------:|:----:|
> | None (w/o attack) | 0.79   | 1.00    |1.00 |    1.00     | 0.95 |
> | **NFPA** | **0.01** | **0.14** | **0.16** | **0.02** | **0.08** |
>
> These results confirm that **NFPA is highly effective in removing conventional image watermarks**, **reducing the average detection rate from 0.95 to 0.08**. This demonstrates strong generalizability of our approach in both synthetic and real-world watermarking scenarios.
>
> ---
>
> We once again thank the reviewer for the valuable feedback and encouragement. Your suggestions have significantly helped improve the **completeness, clarity, and evaluation strength** of our submission. We will incorporate all additional results and analyses in the final version to address your concerns comprehensively.
>
> **References**
>
> [1] Müller et al. *Black-box forgery attacks on semantic watermarks for diffusion models.* CVPR 2025.
> [2] Zhang et al. *The Unreasonable Effectiveness of Deep Features as a Perceptual Metric.* CVPR 2018.
> [a] Wu et al. *Q-ALIGN: Teaching LMMs for Visual Scoring via Discrete Text-defined Levels.* ICML 2024.
> [b] Chen et al. *A Simple Framework for Contrastive Learning of Visual Representations.* ICML 2020.

---

> > ### Comment · Reviewer_m5Bd · 2025-08-02
> >
> > I thank the authors for the rebuttal, which partially alleviated my concerns. After checking comments and rebuttals (including to other reviewers), I would keep my rating unchanged. BTW, LPIPS was a pixel-based metric;  rather, it was computed in the low-dimensional feature space (indicating the semantic and structure information of images).

---

> > > ### Author Response · Authors · 2025-08-02
> > >
> > > Dear Reviewer m5Bd,
> > >
> > > Thank you once again for reviewing our paper and providing valuable feedback. Your insights have greatly helped us enhance the discussion and improve the presentation. We have made every effort to address your concerns in the revised manuscript and have clarified the motivation behind our choice of perceptual metrics. We also appreciate your clarification regarding LPIPS; while we understand it operates in feature space, its sensitivity to pixel arrangement changes may still pose challenges for evaluating our method.
> > >
> > > We would be happy to answer any further questions you may have. Thank you again for your time and thoughtful review.
> > >
> > > Sincerely,
> > >
> > > Authors

---

### Note · Authors · 2025-08-14

We sincerely thank all reviewers for their careful reading, constructive feedback, and recognition of the novelty, quality, and potential impact of our work. We also appreciate the AC for overseeing this thorough and professional review process. Below, we provide a concise summary of our responses, revisions, and clarifications.

### **High-Quality and Novel Contribution**

- **First semantic-level removal attack:** To the best of our knowledge, NFPA represents the first semantic-level attack for image watermark removal, formulated as a next-frame prediction task that extends beyond conventional pixel-level attacks.
- **Practical advantages:** NFPA operates under black-box, zero-shot, data-free, and query-free settings, demonstrating strong practicality and applicability in real-world scenarios.
- **Strong empirical performance:** Across a wide range of watermark types, NFPA effectively removes embedded signals while preserving both visual and semantic fidelity, as validated through multiple quantitative and perceptual metrics.

### **Extensive Validation of NFPA’s Effectiveness**

- **Comprehensive baseline comparisons:** In response to reviewer feedback, we extended experiments to include recent methods (e.g., IRA, CtrlRegen+, Stable Video Diffusion). NFPA consistently achieves higher attack success rates while requiring substantially lower computational resources.
- **Robustness across models and datasets:** NFPA was evaluated on multiple diffusion models (e.g., FLUX.1-dev) and real-world images (MIRFLICKR), demonstrating strong generalization and consistent effectiveness across diverse settings.
- **Semantic and perceptual quality verification:** We incorporated Q-align and CosSim metrics to explicitly quantify semantic fidelity, complementing traditional measures such as FID and CLIP Score. These results confirm that NFPA preserves both perceptual realism and high-level content consistency.

### **Summary**

Through the rebuttal process, we have addressed all major reviewer concerns, including baseline comparisons, computational efficiency, and semantic fidelity. NFPA represents an innovative, robust, and efficient framework for image watermark removal, offering a practical tool for evaluating watermark resilience.

We are committed to ensuring that the revised paper is rigorous and comprehensive, fully addressing all reviewers’ concerns while further enhancing the overall quality and clarity of the work.

---

### Decision · Program_Chairs · 2025-09-17

**Decision:**

Accept (poster)

**Comment:**

The paper proposes a ‘next frame prediction attack’ - phrasing the image watermarking removal task as a next frame prediction i.e. video generation task.

The paper received equivocal reviews initially, but following rebuttal most reviewers shifted to borderline accept (3x BA) and one firm accept (1x A).

m5Bd has concerns over comprehensiveness of baselines and whether the approach worked also for post-hoc watermarks.    The rebuttal added experiments addressing both questions, and these should one included in the final version.  There was another query re: LPIPS but it appears this was due to a technical miscommunication and some improved clarity here in the final paper would be welcome.

seke had several concerns e.g. whether translation i.e. shift would be as effective, and shared concerns over the breadth of comparative evaluation presented.  These were addressed in the rebuttal and should be included in the final version.

g6pS mainly had concerns over the motivation and framing of the task but raised their score post-rebuttal to weak acceptance.

gigz was the most positive reviewer, considering the approach novel with strong validation.

Overall the AC considers this paper a novel approach to watermark attack, that if supported with the additional experiments in the rebuttal, would result in a strong contribution.  The AC notes the consensus view to accept the paper and so also recommends to accept it.